# Diffuse X-ray scattering from correlated motions in a protein crystal

Steve P. Meisburger [1,2], David A. Case[3] & Nozomi Ando [1,2✉]

Protein dynamics are integral to biological function, yet few techniques are sensitive to collective atomic motions. A long-standing goal of X-ray crystallography has been to combine structural information from Bragg diffraction with dynamic information contained in the diffuse scattering background. However, the origin of macromolecular diffuse scattering has been poorly understood, limiting its applicability. We present a finely sampled diffuse scattering map from triclinic lysozyme with unprecedented accuracy and detail, clearly resolving both the inter- and intramolecular correlations. These correlations are studied theoretically using both all-atom molecular dynamics and simple vibrational models. Although lattice dynamics reproduce most of the diffuse pattern, protein internal dynamics, which include hinge-bending motions, are needed to explain the short-ranged correlations revealed by Patterson analysis. These insights lay the groundwork for animating crystal structures with biochemically relevant motions.

[1] Department of Chemistry, Princeton University, Princeton, NJ 08544, USA. [2] Department of Chemistry and Chemical Biology, Cornell University, Ithaca, NY 14850, USA. [3] Department of Chemistry and Chemical Biology, Rutgers University, Piscataway, NJ 08854, USA. ✉email: nozomi.ando@cornell.edu

Conventional structure determination by X-ray crystallography relies on the intense spots recorded in diffraction images, known as Bragg peaks, that represent the average electron density of the unit cell. The average electron density is blurred when atoms are displaced from their average positions, leading to a decay in the Bragg intensities and giving rise to a second signal: a continuous pattern known as diffuse scattering[1,2]. Although the disorder is routinely modeled in structure refinement of Bragg data as atomic displacement parameters (ADPs) or B-factors[3], information about whether groups of atoms move independently or collectively is contained only in the diffuse scattering (Supplementary Fig. 1). However, the diffuse signal is weak compared to Bragg data and challenging to accurately measure. Diffuse scattering has therefore been largely ignored in macromolecular crystallography, and instead, atomic motions have been inferred solely from Bragg data[4–6].

The potential of diffuse scattering as a probe of protein dynamics was envisioned over 30 years ago when Caspar et al.[7] attributed the cloudy diffuse signal from an insulin crystal to liquid-like internal motions. More recently, it has been proposed that diffuse scattering can also disambiguate common structure refinement models that fit collective motions of atoms to ADPs[8]. Motivated by these key ideas, a number of models of protein motion have been proposed to explain macromolecular diffuse scattering[2,9–13]. However, in all cases to-date, agreement between measurement and simulation has been far from compelling[14–20], and thus, the promise of diffuse scattering has not yet been realized.

The main bottleneck in the field has been the lack of accurate data. In particular, the diffuse pattern is typically a small variation on top of a large background and is therefore easily corrupted by intense Bragg peaks. Thus, it has been common practice to heavily process images either by filtering or masking near-Bragg pixels[14,17]. However, this treatment suppresses features that are derived from long-ranged correlations extending beyond the unit cell and may also alter the information contained in the remaining signal. The emerging view is that long-ranged correlations must be considered[2,19,21], but despite the advent of pixel array detectors that are newly enabling[22,23], diffuse scattering data capable of testing such models have not been reported.

To understand the fundamental origins of diffuse scattering from protein crystals, we analyzed the total scattering from the triclinic form of hen lysozyme (Fig. 1a) collected at ambient temperature using a photon-counting pixel array detector (Supplementary Fig. 2A). The triclinic crystals[24] feature low mosaicity and importantly, one protein molecule per unit cell, ensuring that features between the Bragg peaks are fully resolved. By combining high-quality experimental data with new processing methods, we were able to construct a highly detailed map of diffuse scattering without filtering the images. This map reveals, for the first time, a surprisingly large contribution of long-ranged correlated motions across multiple unit cells, while also enabling detection of protein motions in a manner that is consistent with both Bragg diffraction and diffuse scattering.

## Results

**Construction of a three-dimensional reciprocal space map.** For accurate measurements of diffuse scattering at room temperature, the main challenges are to avoid contamination by Bragg peaks and background scattering and to achieve high signal-to-noise while avoiding radiation damage. Using well-collimated and monochromatic synchrotron radiation, we measured the angular broadening (apparent mosaicity) of our triclinic lysozyme crystals to be 0.02–0.03 degrees, which is as small as could be resolved by the diffraction instrument[25]. With such low mosaicity, the sharp, Gaussian-shaped Bragg peaks are readily distinguished from the underlying diffuse scattering (Supplementary Fig. 3A). To take advantage of this low mosaicity, data were collected with fine phi-slicing (0.1 deg). Crystals were held in low-background capillaries (Supplementary Fig. 2A), and low-dose partial datasets were collected from multiple sample volumes. In total, four crystals yielded 5500 images from 11 different sample volumes (Supplementary Fig. 2B, Supplementary Table 1). Using standard crystallography methods, we determined a structure to 1.21 Å (Supplementary Table 2) that agrees well with a previously reported room-temperature structure (PDB ID 4lzt[24], 0.14 Å r.m.s.d.). Analysis of the structure and Bragg intensities shows that radiation damage effects were minimal (Supplementary Fig. 4).

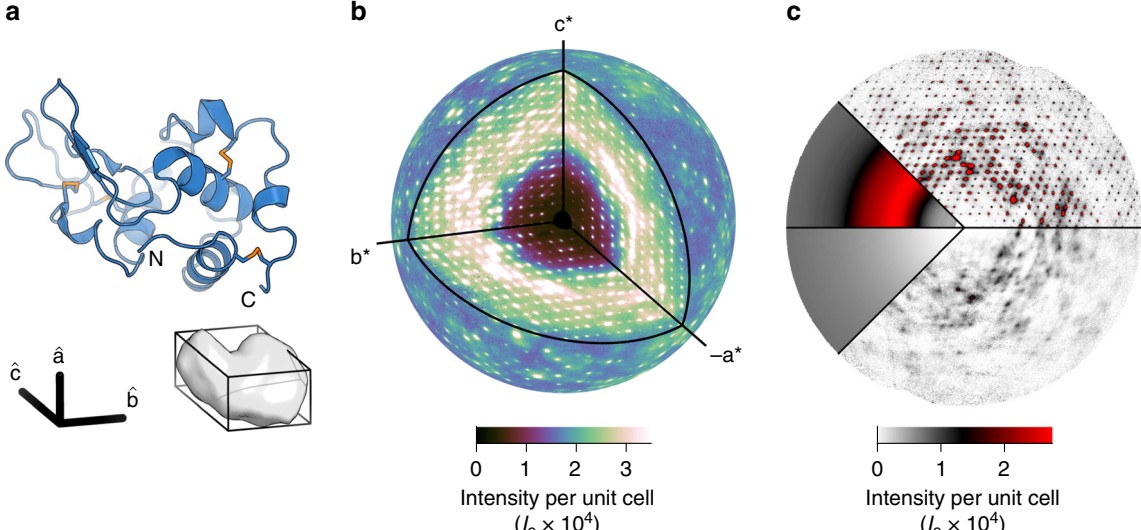

**Fig. 1 Diffuse scattering map of triclinic lysozyme with intensities on an absolute scale of electron units ($I_e$). a** Ribbon diagram of lysozyme (top) and the triclinic unit cell containing one protein (bottom). **b** A highly detailed three-dimensional map of diffuse scattering was obtained. The outer sphere is drawn at 2 Å resolution. **c** The total scattering is made up of three components: inelastic Compton scattering (lower left), a broad isotropic ring that dominates the diffuse signal (upper left), and variational features in the diffuse scattering (right). Intense halos are visible in the layers containing Bragg peaks ($l = 0$ plane, upper right). Cloudy scattering is best visualized in the planes mid-way between the Bragg peaks ($l = 1/2$ plane, lower right).

A three-dimensional diffuse map (Fig. 1b) was constructed from the same set of images (described in detail in the "Methods" section). Background scattering varied with spindle angle (Supplementary Figs. 2A and 5) and was therefore subtracted frame-by-frame (Supplementary Fig. 2C). Scale factors for each image pixel were calculated from first principles to account for X-ray beam polarization, detector absorption efficiency, solid angle, and attenuation by air. Additionally, we utilized the high data redundancy to correct for other experimental artifacts, including self-absorption of the crystal, changes in illuminated volume, differences in efficiency among the detector chips, and excess scattering from the loop and liquid on the surface of the crystal (Supplementary Fig. 6). Each of these corrections improved data quality (Supplementary Fig. 7). The data were accumulated on a fine reciprocal space grid such that the Bragg peaks were entirely contained within the voxels centered on the reciprocal lattice nodes (Supplementary Fig. 3B). In this grid, the reciprocal lattice vectors $\mathbf{a}^*$, $\mathbf{b}^*$, and $\mathbf{c}^*$ are subdivided by 13, 11, and 11, respectively. The map had a maximum resolution of 1.25 Å, and Friedel pairs were averaged, for a total of ~50 million unique voxels.

To enable rigorous comparison between simulations and experiment, we adapted the integral method of Krogh-Moe[26,27] to place the map on an absolute scale of electron units per unit cell (Methods, Supplementary Fig. 8). By doing so, we are able to subtract the inelastic scattering contribution, which depends only on the atomic inventory and is insensitive to molecular structure (Fig. 1c, lower left). The final diffuse map thus represents the coherent scattering of interest (Fig. 1b) with features that depend on structure.

**Phonon-like scattering**. The diffuse scattering is dominated by a broad, isotropic scattering ring with a peak at ~3 Å (Fig. 1c, upper left). Although this ring is generally attributed to water, short-ranged protein disorder also contributes[28,29]. To better visualize the non-isotropic fluctuations, we resampled the full map mid-way between the Bragg peaks and defined the isotropic background as one sigma level below the mean scattering of this map in each resolution bin (Methods, Supplementary Fig. 9). Subtracting this background from the full map reveals clear non-isotropic features, hereafter referred to as "variational" (Fig. 1c, and Supplementary Movie 1). The most striking variational features are the intense halos (Fig. 1c, upper right) that appear to co-localize with Bragg peaks at the reciprocal lattice nodes (Fig. 1b), and are significantly asymmetric in certain directions (Supplementary Fig. 10, left). Overlaid with the halos is a cloudy pattern that is found throughout the map (Fig. 1c, lower right), which we estimate accounts for roughly half of the integrated variational intensity in most resolution bins (Supplementary Fig. 11).

The presence of such intense halo scattering near the Bragg peaks was unexpected, as it implies that the correlations between atoms in different unit cells are significant and long-ranged. In protein crystallography, an outstanding question has been whether such correlations are dynamic in nature, and specifically, due to lattice vibrations[7,9,15,21,28,30]. The scattering intensity of a phonon (vibrational mode) is proportional to the mean squared amplitude of vibration and peaks at certain points in reciprocal space. In particular, a phonon with wavevector $\mathbf{k}$ makes the greatest contribution when the scattering vector $\mathbf{q}$ (with magnitude $|\mathbf{q}| = 2\pi/d$) is parallel to the phonon polarization and displaced from the nearest Bragg peak at $\mathbf{q}_0$ such that $\mathbf{q} - \mathbf{q}_0 = \pm\mathbf{k}$[31]. The scattering of the so-called acoustic phonons, which are thermally excited at room temperature, is proportional to $v_s^{-2}|\mathbf{k}|^{-2}$, where $v_s$ is the speed of sound. Thus, at the Bragg peak locations, acoustic phonon scattering is expected to produce halos with a characteristic $|\mathbf{q} - \mathbf{q}_0|^{-2}$ decay in intensity in any given direction.

With our finely sampled diffuse map, the halo scattering can be inspected directly. We selected three symmetric and intense halos and plotted their intensities along the three reciprocal axes on a double-log scale, where a power law is a straight line (Fig. 2a, left). Both the power-law behavior and the characteristic exponent are fully consistent with acoustic phonon scattering. Furthermore, the fact that the plot remains linear as $\mathbf{q}$ approaches $\mathbf{q}_0$ implies that the lattice vibrations are coherent over at least $2\pi/|\mathbf{k}_{\min}| \sim 300$ Å or ~10 unit cells. The characteristic exponent of approximately $-2$ is also found for other intense halos throughout the map (Fig. 2a, right). These results are highly suggestive of vibrational lattice dynamics.

**All-atom molecular dynamics simulations**. Although all-atom MD simulations have previously been used to investigate the contribution of protein dynamics to diffuse scattering[12,16,29,32–34], the effect of long-ranged correlations due to lattice disorder has not been examined. We thus performed all-atom MD simulations of triclinic lysozyme crystals as a function of supercell size (Methods). Experimentally determined coordinates were used to define and initialize an array of proteins comprising the supercell, and periodic boundary conditions were imposed to remove edge effects. Supercells composed of 1, 27 ($3 \times 3 \times 3$), 125 ($5 \times 5 \times 5$), and 343 ($7 \times 7 \times 7$) unit cells were simulated for 5, 5, 2, and 1 μs, respectively. Guinier's equation[35] was used to calculate the diffuse intensity per unit cell from the simulation trajectory (Methods). Because the boundary conditions are periodic, the diffuse scattering was sampled at integer subdivisions of the reciprocal lattice (i.e., the number of unit cells in each direction).

In the 1 unit-cell simulation (Fig. 2b), cloudy variational features are observed in rough qualitative agreement with the experiment (Fig. 1c, Supplementary Fig. 9B, D), suggesting that local protein and solvent dynamics contribute to the observed diffuse scattering. Unlike simpler models that do not include liquid correlations in the bulk solvent, MD provides a prediction for the isotropic component (Supplementary Fig. 9C). The overall correlation of the isotropic component is 0.9965 between 25 and 1.25 Å resolution, and the magnitude is also similar (Supplementary Fig. 9A, C). However, halos are absent, consistent with the lack of intermolecular disorder enforced by a 1 unit-cell simulation. As the size of the supercell is increased, the diffuse scattering pattern evolves in a complex manner with the halos becoming increasingly apparent (Fig. 2b), confirming that they depend on intermolecular correlations and lattice degrees of freedom. In the 343 unit-cell simulation, the r.m.s. displacement of each chain about its center of mass was 0.20–0.22 Å in each direction. Although this may seem to be a small motion, the intense halo signal is derived from constructive interference of scattered radiation from many proteins moving collectively.

In the 343 unit-cell simulation (Fig. 2b), the simulated scattering contains both cloudy and halo features similar to those observed experimentally. However, the MD does not reproduce the experiment on an absolute scale (Fig. 2c, black diamonds). To make a quantitative comparison, we interpolated the experimental map on the simulation grid ($7 \times 7 \times 7$) and computed the Pearson correlation coefficient (CC) between the two in thin shells of constant resolution (Fig. 2d, orange). Although we obtain a reasonable CC of ~0.7 up to 2 Å resolution, the CC decreases at higher resolution. Moreover, there is a significant gap between CC (Fig. 2d, orange) and CC* (Fig. 2d, black dashed), which estimates the maximum CC a model can achieve, given the precision of the data[36]. This discrepancy indicates that model-data agreement is not limited by noise and instead points to shortcomings of the crystal model, including the current MD force fields. In particular, the accuracy of MD for diffuse

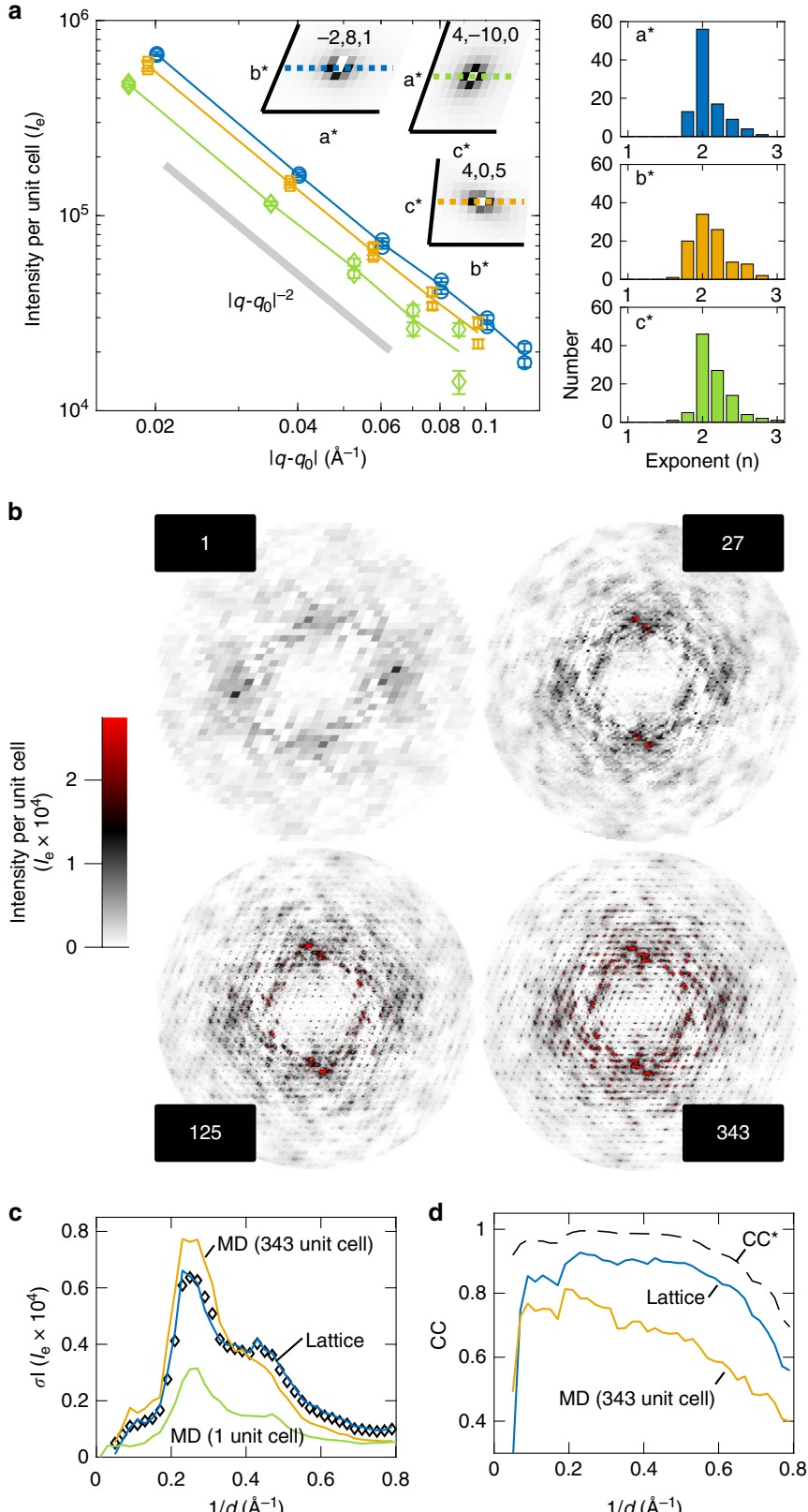

scattering appears to be limited by errors in the average electron density (Supplementary Fig. 12). To gain insight into the underlying physics of the variational scattering features, we thus sought simpler dynamical models that can be refined to fit both the Bragg and diffuse data.

**Lattice dynamics refined against diffuse scattering.** Given the evidence for acoustic phonon scattering, we investigated whether vibrational models can capture the observed halo shapes and intensities. We developed a lattice dynamics model where each protein is able to move as a rigid body that is connected to

**Fig. 2 Evidence for long-ranged correlations in experimental maps and molecular dynamics (MD) simulations. a** Throughout the diffuse map, intense halo scattering is observed around Bragg reflections. Halo profiles centered on three Bragg reflections ($\mathbf{q}_0$) show a power-law decay with an exponent close to $-2$ (gray line) along the directions ($\mathbf{q} - \mathbf{q}_0$): $\mathbf{a}^\star$ (blue), $\mathbf{b}^\star$ (orange), and $\mathbf{c}^\star$ (green). Error bars represent the standard error of the mean. Histograms of the best-fit exponent along $\mathbf{a}^\star$, $\mathbf{b}^\star$ and $\mathbf{c}^\star$ (top to bottom) for the 100 most-intense halos between 2 and 10 Å resolution also show that $-2$ is the most frequent value. **b** Halo features appear in simulated scattering from supercell MD as the simulation size is increased from 1 to 343 ($7 \times 7 \times 7$) unit cells. Each panel shows the variational component in the $l = 0$ plane. **c** Although increasing the supercell size improves agreement (green to orange), MD does not reproduce experiment on an absolute scale (black diamonds), as judged by the standard deviation profile of the diffuse intensity. In contrast, much better agreement is obtained with the lattice model described in Fig. 3 (blue). **d** MD displays a worse correlation (CC) with experiment (orange) compared to the lattice model (blue). The dashed line represents theoretical limit of the experimental data, CC*.

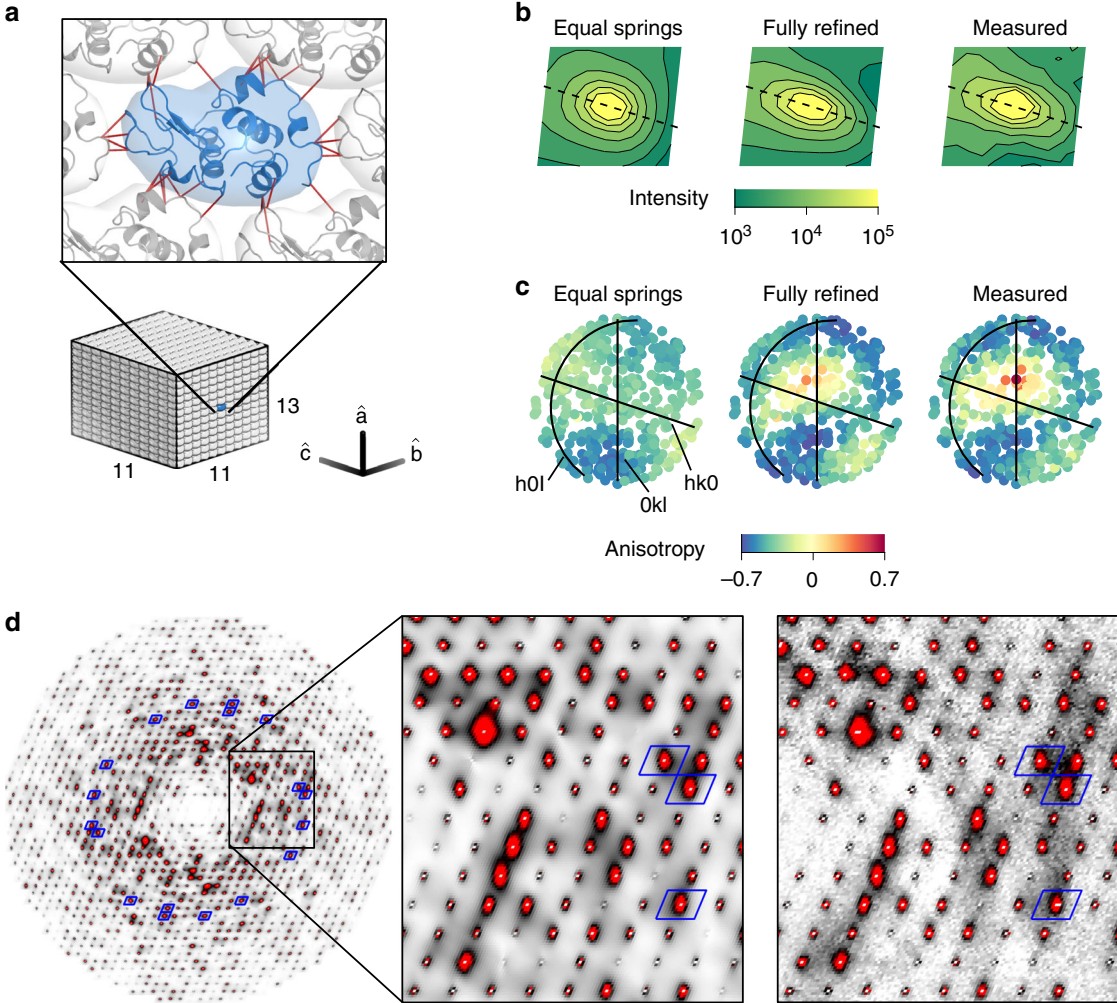

**Fig. 3 Lattice dynamics model refined to diffuse scattering. a** A lattice dynamics model was constructed with rigid protein units arranged in a $13 \times 11 \times 11$ supercell with a linear combination of Gaussian and directional springs connecting the $C_\alpha$ atoms of residues involved in lattice contacts (dark red lines). Spring constants were refined to fit the variational scattering around 400 intense Bragg peaks between 2 and 2.5 Å resolution. **b** Comparison of predicted and measured halo intensity around the (1,2,13) Bragg reflection in the $h = 1$ plane. The plane perpendicular to the scattering vector is indicated by a dashed line. The model with equal Gaussian springs does not reproduce this shape as well as the fully-refined model. **c** The shape anisotropy of each of the 400 halos used for model fitting was quantified and mapped as an equal-area projection of the hemisphere centered on $\mathbf{b}^\star$. Full refinement of the spring constants was needed to reproduce the pattern of halo anisotropy seen in experiment. **d** The simulated one-phonon scattering for the fully-refined lattice model (left) is compared with the measured variational scattering (right) in the $l = 0$ plane. The intensity scale is the same as Fig. 1c. Blue boxes surround halos that were included in the fit.

neighboring molecules via spring-like interactions (detailed in Methods). The proteins were arranged in a $13 \times 11 \times 11$ supercell to match the sampling of the experimental map (Fig. 3a). Residues of neighboring molecules that form lattice contacts were linked by a pair potential between alpha carbons (Fig. 3a, dark red lines), reflecting a restoring force that depends on the relative displacements of the two end-points. For generality, we allowed each pair

potential to be a linear combination of two types of springs: Gaussian and directional. Gaussian springs[37] have a restoring force that is independent of the direction of the displacement relative to the spring, and directional springs[38] have a restoring force only along the vector between the end-points.

The model was refined against a set of 400 intense halos between 2 and 2.5 Å resolution, consisting of a total of ~600,000 voxels. As

there are three-dimensional halos associated with all 30,108 unique Bragg reflections, these 400 represent a small subset (1.3%). The spring constants were initially restrained to be all Gaussian and equal, and restraints were relaxed during subsequent stages of refinement. For a given set of springs, the equations of motion were solved by the Born/Von-Karman method[31,39,40], and the diffuse scattering was calculated using the one-phonon approximation (detailed in Methods). At each refinement stage, we monitored the overall $\chi^2$ value between the experimental and simulated scattering (Supplementary Fig. 13A), as well as the ability of the model to reproduce the halo shape (Fig. 3b). To monitor agreement with halo anisotropy, we fit each of the halos to a function of the form $I = [(\mathbf{q} - \mathbf{q}_0)^{\mathrm{T}} \mathbf{G} (\mathbf{q} - \mathbf{q}_0)]^{-1}$, where $\mathbf{G}$ is a $3 \times 3$ positive definite matrix, and defined an anisotropy parameter, $a = G_\perp / G_\parallel - 1$, where $G_\parallel$ is the component of $\mathbf{G}$ parallel to $\mathbf{q}_0$, and $G_\perp$ is the average of the perpendicular components. The fully-parameterized model was necessary to reproduce the pattern of halo anisotropy (Fig. 3c, Supplementary Fig. 13B).

After refining the lattice dynamics model using the working set of 400 halos (Fig. 3d, blue boxes), we simulated the complete diffuse scattering map over the full resolution range. Remarkably, the simulation reproduces many of the variational scattering features observed in experiment (Fig. 3d, right). Anisotropic halo shapes are reproduced even in regions of the map that were not used to refine the model (Fig. 3d, regions outside of blue boxes). Streaks in the pattern are also reproduced and can be attributed to a modulation of the halos by the molecular transform (Supplementary Fig. 10). Moreover, we find that the halos do not decay to zero mid-way between the Bragg peaks as previously expected[9], giving rise to a cloudy pattern that resembles the cloudy variational scattering in the data (Fig. 3d, right). The standard deviations of intensity have very similar profiles and absolute magnitudes (Fig. 2c, blue solid and diamonds), suggesting that the lattice dynamics make the most significant contribution to the scattering variations. This conclusion is supported by the much smaller variations seen in the 1 unit-cell MD simulation (Fig. 2c, green), where lattice disorder is absent by construction.

As before, the agreement between the experimental and simulated maps was assessed with CC and CC*. For the lattice dynamics model, the CC is excellent in regions of high signal-to-noise (CC ~ 0.9 between 2 and 5 Å resolution) (Supplementary Fig. 14, solid) and only limited by the experimental precision at higher resolution (Supplementary Fig. 14, dashed). To improve the signal-to-noise, the maps were interpolated on a $7 \times 7 \times 7$ grid (Fig. 2d, blue), enabling direct comparison with the 343 unit-cell MD simulation (Fig. 2d, orange). Strikingly, the lattice dynamics model clearly outperforms all-atom MD in its ability to describe the variational component (Fig. 2c, d).

The lattice model can be further assessed against existing biophysical data. Our model predicts that sound waves should propagate through the crystal (Supplementary Movie 2). Based on the calculated dispersion relations of the acoustic vibrational modes (Supplementary Fig. 15), we obtain longitudinal sound velocities of 1.0–1.3 km s$^{-1}$ and corresponding transverse velocities that are slower by a factor of 1.3–2.1 depending on the propagation direction (Supplementary Table 3). Although few measurements of sound propagation have been made in protein crystals, longitudinal velocities have generally been reported to be ~2 km s$^{-1}$ [41–43], and transverse velocities are estimated to be 2–3 times slower[41,44]. Thus, our interpretation that the halo scattering arises from dynamic, rather than static, disorder appears physically reasonable.

**Contribution of lattice dynamics to atomic motion**. As described earlier, the amount of apparent motion for each atom

can be quantified from Bragg data by refining individual ADPs, the 6 components needed to describe a 3-dimensional Gaussian probability distribution. Our data quality was sufficient to refine full anisotropic ADPs for every non-H atom. To determine the extent to which lattice dynamics contribute to atomic motion, corresponding ADPs were calculated directly from the refined lattice model (Methods, Supplementary Table 4). In Fig. 4a, the full ADPs of the backbone atoms are reduced to a single isotropic B-factor per residue to facilitate visual comparison. Overall, the backbone B-factors for the lattice model (5.2 Å$^2$ on average) fall below those of experiment (9.4 Å$^2$ on average). The B-factors from the lattice model show small variations, which can be attributed to rigid-body rotational motion with an r.m.s. amplitude of 0.8° (Supplementary Table 4). However, the B-factor variations in the data are much more pronounced (Fig. 4a), particularly for side-chains (Supplementary Fig. 16A). These residual B-factors imply the existence of internal dynamics, in other words, that atoms within the protein undergo collective motions.

**Protein dynamics refined against Bragg data**. The collective motions of lysozyme have been a topic of long-standing biophysical interest since hinge-bending motions between the two domains (Fig. 4b, blue and green) were first proposed as a mechanism for substrate binding and release[45,46]. To investigate the presence of such collective motions, we developed an elastic network model, in which each protein residue moves as a rigid body, and all non-H atoms within 4 Å are coupled with directional springs (Methods). As with the lattice model, the crystal environment was modeled with intermolecular springs and periodic boundary conditions, and the dynamics were calculated using the Born/Von-Karman method. In order to model only the internal protein dynamics, the Hessian matrix describing the restoring forces was modified to suppress rigid-body motion of the entire protein. The model was parametrized with one coupling constant per residue (i.e., 129 free parameters total) so that springs joining a residue pair were assigned a spring constant equal to the geometric mean of the coupling constants (Methods). The parameters were then refined by minimizing the least-squares difference between all components of the calculated (lattice + internal) and experimental ADPs derived from Bragg data.

The refined model is able to reproduce the pattern of B-factors obtained experimentally (Fig. 4a and Supplementary Fig. 16A, B). To assess the importance of hinge-bending in the model, we examined the covariance matrices $\mathbf{C}_{ij}$ for all alpha carbon pairs and calculated a "directional correlation", which is the component of $\mathbf{C}_{ij}$ along the inter-atomic vector normalized by the r.m.s. displacements of the two atoms (Methods). By this measure (Fig. 4c), the two domains are significantly anti-correlated as expected for hinge-bending motion.

**Contribution of protein dynamics to diffuse scattering**. Lattice dynamics account for the bulk of the variational diffuse scattering, as evaluated by CC and standard deviation (Fig. 2c, d). However, these statistics emphasize the most intense features in the signal, which in this case are the halos. To assess the more subtle contributions of internal protein motions, correlations in the signal should be separated based on length-scale. We thus calculated the diffuse Patterson (also known as 3D-ΔPDF), which is the Fourier transform of the diffuse scattering. The diffuse Patterson map represents the mean autocorrelation of the difference electron density, $\Delta\rho = \rho - \langle\rho\rangle$, such that a vector from the origin of the map corresponds to a vector between two points

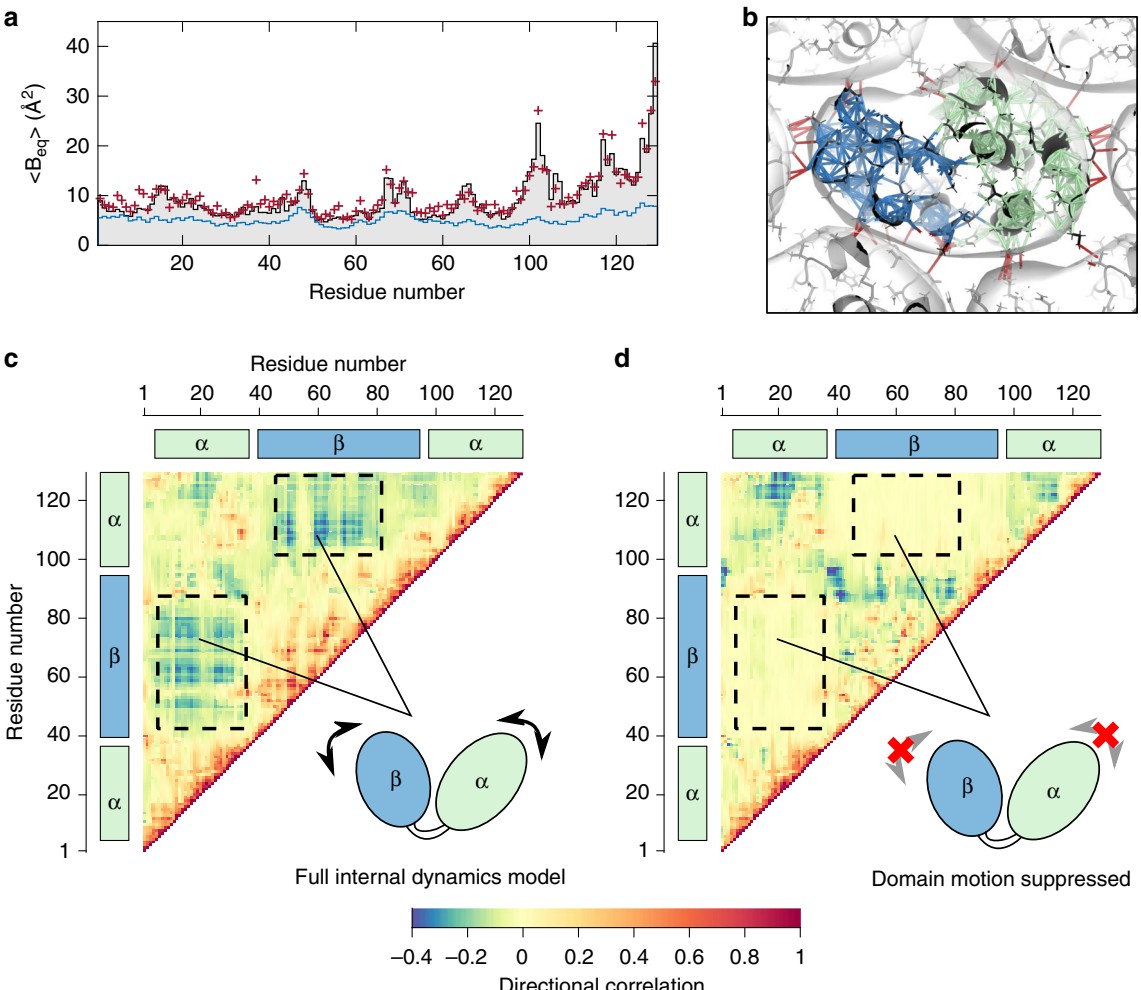

**Fig. 4 Models of collective internal motions in lysozyme refined to Bragg data. a** Apparent atomic motions can be evaluated by comparing the atomic displacement parameters (ADPs) obtained experimentally from Bragg data with those calculated from models. To facilitate visual comparison, the ADPs of backbone atoms are averaged to produce a single isotropic B-factor per residue. The lattice dynamics model in Fig. 3 (blue curve) underestimates the experimental B-factors (gray bars), but a good fit is obtained by combining lattice dynamics with the internal dynamics described in (**b**, **c**) (dark red symbols). **b** The model for internal dynamics was constructed using an elastic network with rigid residues. Both intermolecular (dark red lines) and intramolecular contacts (blue and green lines, corresponding to the $\alpha$ and $\beta$ domains, respectively) were modeled as springs, and the spring constants were refined to fit the residual ADPs, i.e. the experimental ADPs that are unaccounted for by lattice dynamics. **c** In the full internal dynamics model, the $C_\alpha$ atoms in the $\alpha$ and $\beta$ domains show negative directional correlations (dashed boxes), indicating that their motions are anti-correlated and consistent with hinge-bending. **d** The two domains have no correlations when their motions are suppressed in the model refinement.

in the crystal. Thus, the central part of the diffuse Patterson is affected only by those correlations that are short-ranged.

At large distances, the experimental diffuse Patterson displays peaks at the lattice nodes as expected (the Fourier transform of a lattice is also a lattice) (Fig. 5a, left), whereas continuous features are most intense at short distances (Fig. 5a, right). To determine whether lattice dynamics alone can account for the short-ranged correlations, the diffuse Patterson was calculated directly from the refined lattice model (Methods). Although the simulated and experimental maps share similar features (Fig. 5a, b), the amplitudes of the fluctuations are clearly underestimated for distances shorter than ~10 Å (Fig. 5f, blue curve vs. diamonds).

In contrast, the diffuse Patterson calculated from the internal motion model, refined to the residual ADPs, shows prominent fluctuations for pair distances less than ~10 Å but very little outside this range (Fig. 5c, f, green). Assuming that the protein internal motions are independent of lattice motions, the diffuse Patterson maps can be added (Fig. 5d). The combined model displays remarkable agreement with the experimental map and

reproduces the characteristic decay of fluctuation amplitude almost exactly (Fig. 5f, dark red curve vs. diamonds). To assess the agreement more quantitatively, the CC profile was calculated in reciprocal space (a Fourier transform of the $2 < r < 25$ Å region). The combined model (Fig. 5g, h, dark red) displays a significant gain in CC over the lattice model alone (Fig. 5g, h, blue). The level of model-data agreement that we obtain is excellent (Fig. 5g, dark red), especially when compared to the all-atom MD simulation (Fig. 5e, g, orange) as well as all previously reported studies[14,15,17–20,33].

The question of model quality has consequence to protein crystallography, where it is common practice to fit models of collective motion to the B-factors, since this often increases the data-to-parameter ratio. Diffuse scattering has been proposed as a means of critically evaluating these models[8]. To explore this idea, we repeated refinement of the internal elastic network model with domain motions selectively suppressed. This restrained model has the same number of free parameters as the unrestrained model (Supplementary Note 1) and it is also able to reproduce the

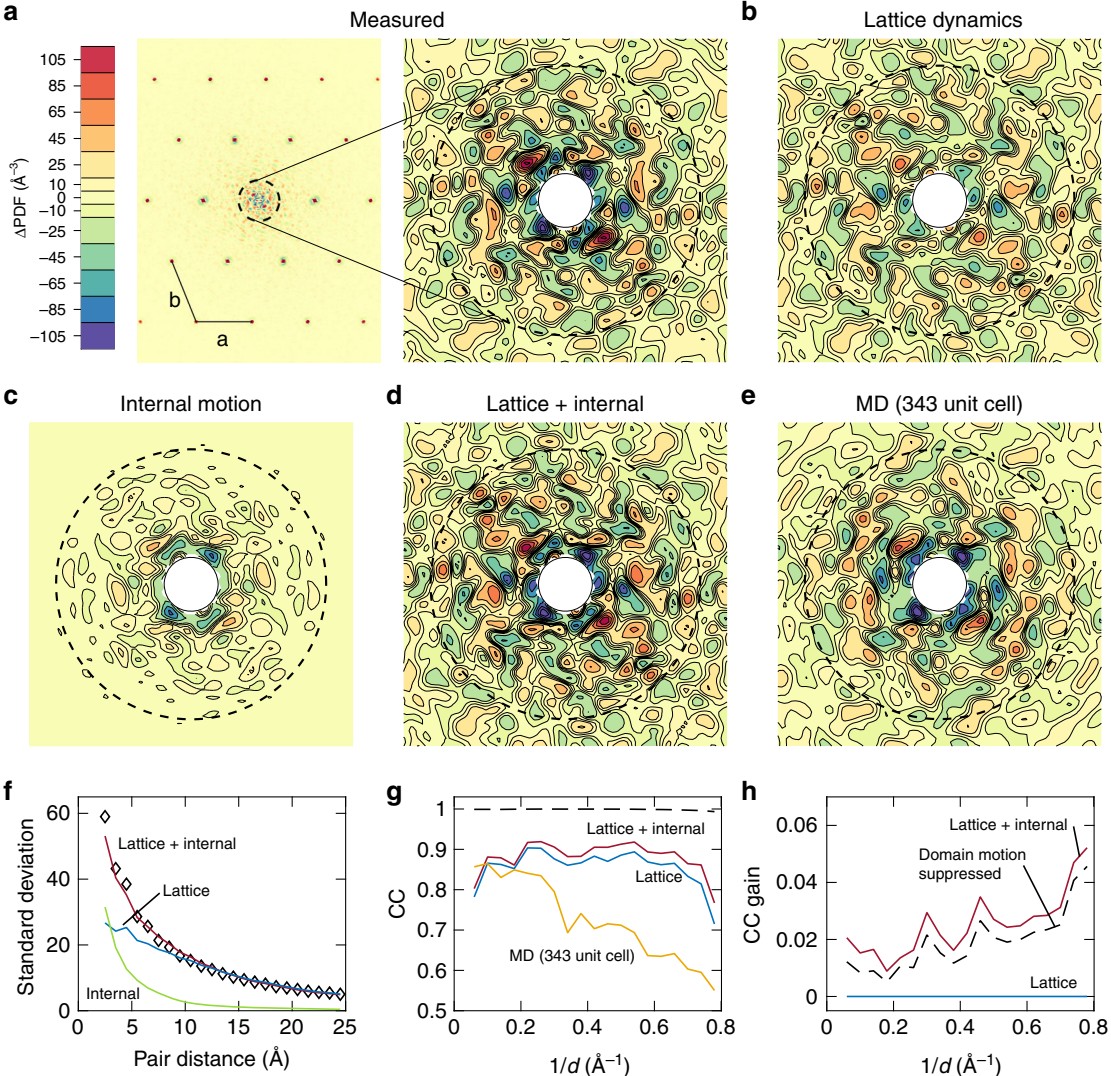

**Fig. 5 Detection of internal motions by diffuse Patterson analysis. a** The diffuse Patterson map represents the autocorrelation of the difference electron density and is a function of the vector **r** between points in the crystal (dashed circle corresponds to |**r**| = 10 Å). The experimental diffuse Patterson in the a-b plane contains peaks at the lattice nodes (left) and continuous fluctuations that are most intense near the origin (right). **b–e** Diffuse Patterson maps simulated from models. The lattice model underestimates the fluctuations at short length scales, but addition of full internal dynamics reproduces the experimental pattern. **f** The standard deviation of the diffuse Patterson maps in spherical shells of constant pair-distance for the experimental map (black diamonds), lattice model (blue), internal model (green), and the combined model (dark red). **g** The reciprocal space correlation coefficient (CC) between experiment and simulation in shells of constant resolution within the central part of the Patterson map (2 < |**r**| < 25 Å) . The colors are the same as in (**f**). The 343-unit cell MD simulation is in orange. The dashed line shows CC*. **h** Gain in CC relative to the lattice dynamics model alone (blue) for the combined model (dark red) and the model in which domain motion was suppressed (black dashed line).

experimentally derived B-factors ($R^2 = 0.88$ for both models, see Supplementary Fig. 16C). However, the internal dynamics are significantly different (Fig. 4d), underscoring the challenge of distinguishing differing models of protein motion from Bragg data alone. Yet, the two models are distinguishable by diffuse scattering: fluctuations in the diffuse Patterson decay more rapidly with domain motions suppressed (Supplementary Fig. 17), leading to a subtle but systematically worse CC, particularly at high resolution (Fig. 5h, dashed).

## Discussion

By studying the total X-ray scattering from triclinic lysozyme crystals both experimentally and theoretically, we were able to obtain fundamental insight into the collective motions that produce macromolecular diffuse scattering. Simple vibrational models of the lattice and internal dynamics were developed that explain the electron density correlations spanning two orders of magnitude in length-scale. Vibrations of the entire protein in the lattice account for the shapes and magnitudes of the diffuse halo features and about half of the backbone ADPs, while internal motions of the protein make up the remainder. The collective nature of these internal motions was investigated by diffuse Patterson analysis, which separates correlations based on the inter-atomic vector. Remarkably, we found that two models that fit the ADPs equally well could be distinguished by their agreement to the experimental diffuse Patterson, experimentally demonstrating a key application of diffuse scattering proposed a decade ago[8]. Finally, although the MD was limited in its ability to reproduce the variational diffuse scattering, our results demonstrate that this signal provides an excellent experimental benchmark for improving simulations in the future.

For over 30 years, the ultimate goal of diffuse scattering studies has been to capture internal protein motions from crystallographic data. The success of previous efforts has been limited primarily by data quality as well as the assumption that the variational scattering is largely due to internal motion. In fact, lattice disorder contributes significantly, further underscoring the need for detailed, high-quality data and realistic models. Despite the added challenges, we have also shown that by accounting for lattice dynamics, the remaining diffuse signal indeed contains information about internal motion and can be used to differentiate alternate models. With the initial goal of the diffuse scattering field realized, the next grand challenge of refining structural models that are consistent with the total scattering now appears within reach.

## Methods

**Crystallization**. Triclinic crystals of lysozyme were obtained by the micro-batch method with temperature-cycling to select this crystal form based on its phase diagram[47]. Lyophilized hen egg white lysozyme (Hampton Research) was dissolved in 20 mM sodium acetate (NaOAc) pH 4.6 at a stock concentration of 100 mg per mL, passed through a 0.2 μm filter, and used without further purification. All other reagents were purchased from Sigma, unless noted. Triclinic crystals were grown using a microbatch-under-oil technique with 8 μL drops containing 5–15 mg per mL protein, 224–300 mM NaNO₃, and 50 mM NaOAc pH 4.5 that were covered with 20 μL paraffin oil (Hampton Research). Crystallization trays were set up at room temperature, moved to 4 °C for 8–12 h, and then returned to room temperature. Both triclinic and monoclinic crystals nucleate during the 4 °C incubation, however, after returning to room temperature the triclinic crystals grow at the expense of the monoclinic form[47].

**Data collection**. X-ray data were collected using the macromolecular crystallography beamline F1 at the Cornell High Energy Synchrotron Source (CHESS), which provided a 12.693 keV X-ray beam collimated to 0.1 mm diameter. Room-temperature data collection was performed using the plastic capillary sheathing method[48]. Crystals were harvested using low-scatter kapton loops (MicroLoops, MiTeGen), taking care to minimize the amount of solvent surrounding the crystal, and placed within 2 mm diameter, 25 μm wall poly(ethylene terephthalate) capillaries (MicroRT, MiTeGen) with 10 μL reservoir solution in the tip. During X-ray exposure, images were recorded every 0.1° using a pixel-array detector (Pilatus3 6M, Dectris) while rotating the sample at 1° s⁻¹. A dose rate of 1.3 kGy s⁻¹ was estimated assuming a flux of $2.5 \times 10^{10}$ photons s⁻¹ and a mass energy-absorption coefficient[49] of $\mu_{en}/\rho = 2.0$ cm² g⁻¹. After 50 s of exposure (~65 kGy), the sample was refreshed by translating to a new spot or replacing the crystal. A background dataset was collected for each crystal by translating the sample out of the beam along the spindle axis and collecting 1 s exposures while rotating at 1° s⁻¹.

**Structure determination from Bragg data**. Bragg data were integrated using xds[50], with geometric parameters refined at 2° increments. The fitted peak profiles and mosaicity per frame were examined to verify that the crystal had not slipped or cracked. The best wedges were then scaled and merged using aimless[51] (Supplementary Table 1). Model building and refinement were carried out using the ccp4 suite of programs[52,53]. The initial model was prepared from PDB ID 4lzt[24], using the most probable (highest occupancy) protein atom coordinates only. Structure refinement was carried out using alternate runs of REFMAC5[54] and manual modeling in coot[55], with atomic displacement parameters included in the final rounds. Alternate conformers were modeled when justified by the electron density and stereochemistry. The atomic coordinates and structure factors have been deposited in the Protein Data Bank under accession code 6o2h. Data collection and refinement statistics are shown in Supplementary Table 2.

**Overview of diffuse data processing**. Reciprocal space maps were generated in Matlab (The Mathworks) as described in the following sections. Briefly, an integration mask was first produced to separate rapidly varying features (including Bragg peaks) from continuously varying features (including diffuse scattering) in three-dimensional reciprocal space. Following per-pixel image corrections, the integration mask was used to generate a coarse continuous scattering map and a Bragg map. Using the coarse continuous scattering map, a scaling model was refined to globally minimize the discrepancy between redundant observations. Bragg intensities were corrected and brought into agreement with the values used for structure determination in the previous section. The intensities were then placed on an absolute scale. Finally, continuous scattering intensities were accumulated on a fine grid to produce the final diffuse map. The scaling corrections from the previous step were applied during integration, and redundant observations were merged without further scaling. A detailed description of each operation is below.

**Construction of integration mask**. As the Bragg intensities and continuous scattering require different corrections, a sensitive moving-window filter was first used to detect and mask out rapidly varying features. The filter algorithm compared the observed count distribution to that expected from Poisson statistics, as described below. Briefly, a voxel was masked out if its exclusion made the neighborhoods to which it contributes more Poisson-like according to the Kullback-Leibler (KL) divergence of the observed and ideal distributions. The unmasked voxels then describe a function that varies smoothly on the scale of the reciprocal space grid.

X-ray images were processed in 2° wedges. Each pixel was mapped onto a provisional reciprocal unit cell, where the reciprocal unit cell was subdivided by a factor of 5 in each direction. For each voxel, a histogram of counts per pixel was accumulated. Using these count histograms, the filtering algorithm proceeded as follows. The neighborhood (filter window) was defined as the set of voxels within a Euclidian distance of ≤2 grid units from the central voxel. For each neighborhood, the weighted median count rate $r_{median}$ was found, as well as the KL divergence of the total count histogram from the expected Poisson distribution with rate $r_{median}$. A voxel was masked if its exclusion reduced the sum of KL divergences for all neighborhoods. First, the voxels were ranked by this change in KL divergences, $\Delta_{KL}$, in ascending order (worst offenders first). Then, voxels were masked progressively, and the $\Delta_{KL}$ values of neighboring voxels were updated without re-sorting. The algorithm halted when encountering a voxel with $\Delta_{KL} \geq 0$. In the resulting integration mask, the unmasked voxel grid represented the continuous scattering, whereas the masked voxel grid consisted mainly of Bragg peaks.

**Integration and scaling**. Per-pixel image corrections were applied prior to integration. Scale factors for each image pixel were calculated from first principles to account for X-ray beam polarization, detector absorption efficiency, solid angle, and attenuation by air (see Section 2 of the Supplementary Methods). The background count rate for each pixel was estimated from an exposure where the crystal was translated out of the beam along the spindle axis (Supplementary Fig. 2C).

Using the mask generated in the previous step, the unmasked and masked voxels were then integrated separately in 2° wedges. To generate a coarse map of continuous scattering, the unmasked voxel grid was reduced to one sample per reciprocal lattice node. In addition, observations of the same voxel in adjacent wedges were combined. Then, the geometric and background corrections were applied to the photon counts to generate a map of $I_{meas}$ for the continuous scattering (Equation 32 in the Supplementary Methods). The masked voxels containing Bragg peaks were integrated in a similar manner, except that the local diffuse background was subtracted and the Lorentz correction was applied (Equation 34 in the Supplementary Methods). For the background, the value of $I_{meas}$ for the coarse continuous scattering map was used. The Bragg intensities were further filtered to remove partial observations. The total reciprocal space volume sampled by the detector during integration (the accumulation over contributing pixels of Equation 28 in the Supplementary Methods) was compared with the actual volume of the masked voxels. The peak was considered to be fully recorded if the volumes agreed within 5%. This rejects a large fraction of the recorded Bragg peaks, however, they are later replaced using more precise integration methods, described below.

Using the coarse continuous scattering map, a scaling model was refined in order to minimize the discrepancy of redundant observations and correct for experimental artifacts. In this case, redundancy comes from Friedel symmetry and the fact that different wedges of data overlapped in reciprocal space. The scaling model related the expected intensity of an observation $i$ to the merged intensity $I_{merge}$, in terms of four correction factors, as follows:

$$I_{pred}(i) = a(x_i, y_i, \phi_i)d(p_i)\Big[b(\phi_i)I_{merge}(\mathbf{h}_i) + c(s_i, \phi_i)\Big], \quad (1)$$

where $I_{pred}$ is the model's prediction for the measured intensity, $\mathbf{h}_i$ is the index of the symmetry-equivalent reflection in the asymmetric unit of reciprocal space, and $a$, $b$, $c$, and $d$ are functions of the experimental geometry; $\phi_i$ is the spindle rotation angle, $s_i = |\mathbf{s}_i|$ is the scattering vector magnitude, $p_i$ is the detector chip index, and $(x_i, y_i)$ is the position in the detector plane. Roughly speaking, $a$ corrects for absorption, $b$ corrects for overall changes in illuminated volume and beam intensity, $c$ is strictly positive and corrects for excess isotropic scattering (which may occur if extra material, such as the sample loop, passes through the beam), and $d$ corrects for detector chip efficiency (flat-field errors). The continuous functions $a$, $b$, and $c$ were obtained by linear interpolation on multi-dimensional grids. A $9 \times 9$ grid was used for the detector plane position, 100 grid points were used for scattering vector ($0 < s < 0.9132$ Å⁻¹), and 26 grid points were used for the spindle angle coordinate of each 50° data wedge. A set of 960 discrete values was used for $d$, corresponding to the 960 detector chips in the Pilatus 6M.

The parameters of the scaling model were fit by minimizing the sum of the $\chi^2$ and regularization terms, as follows:

$$\mathcal{H} = \sum_i \Big(I_{meas}(i) - I_{pred}(i)\Big)^2 \sigma_i^{-2} + \sum_j \lambda_j \mathcal{B}_j, \quad (2)$$

where $\sigma_i$ is the uncertainty (standard error) estimate for $I_{meas}(i)$, $\mathcal{B}_j$ are the regularization functions and $\lambda_j$ are the corresponding weights (Lagrange multipliers). The regularization functions are used to stabilize refinement and to

enforce smoothness of the correction factors. For the correction factors $a$, $b$, and $c$, smoothness was enforced by minimizing the second derivative. Discrete approximations[56] of the following integrals were used: $\int d\phi dx dy \left| \partial_\phi^2 a \right|^2$, $\int d\phi dx dy \left| \partial_x^2 a + \partial_y^2 a \right|^2$, $\int d\phi \left| \partial_\phi^2 b \right|^2$, $\int d\phi ds \left| \partial_\phi^2 c \right|^2$, $\int d\phi ds \left| \partial_s^2 c \right|^2$. In addition, the offset correction was forced to be positive, and to stabilize the refinement, its magnitude was minimized using a discrete approximation of $\int d\phi ds \left| c \right|^2$. Finally, the detector correction factors were regularized using $\sum_p \left| d(p) - 1 \right|^2$, which ensures $d = 1$ in the absence of data. The nonlinear minimization problem was solved iteratively by alternately minimizing $\mathcal{H}$ with constant $I_{merge}$ (a linear problem) and updating $I_{merge}$ given the new scale factors[57]. To simplify the implementation, each set of parameters was refined individually (or in pairs) with the others held fixed. Satisfactory results were obtained by refining corrections in the following sequence: $\{b, o, cb, o, c, a, d\}$, where $cb$ refers to fitting the model for $c$ followed by $b$, and $o$ is an outlier rejection step (Supplementary Figs. 6, 7).

After refining the scaling model, redundant observations in the continuous scattering map were merged. Observations more than $5\sigma$ from the mean were excluded. The estimated Bragg intensities, obtained from integration of the masked voxels, were also scaled and merged using the same model, except that the offset correction was omitted and an outlier cutoff of $2\sigma$ was used. The merged values were compared with the Bragg intensities integrated and merged using xds[50] and aimless[51]. A single scale factor was found to bring the xds/aimless values into agreement with our Bragg intensity map. Since the intensities determined by xds and aimless are more accurate and complete than our estimates, the xds/aimless values were used instead for all subsequent analysis. Doing so also ensures that the Bragg intensities matched the values used for structure determination.

**Placement of intensities on an absolute scale.** After merging, the intensities were placed on an absolute scale. The overall scale factor was found by adaptation of the total intensity method originally described by Krogh-Moe (K-M)[26,27]. The standard K-M method, described in Section 2.3 of the Supplementary Methods, involves predicting the contribution of individual atoms to the total intensity ($I_{total,predicted}$, Equation 40 in the Supplementary Methods) and comparing the prediction to the measured value ($I_{total,measured}$, Equation 41 in the Supplementary Methods) to determine a scale factor $\alpha$ as follows:

$$\alpha = \frac{I_{total,predicted}}{I_{total,measured}}. \tag{3}$$

To test for convergence, the scaling factor was calculated in two ways: first using the standard K-M method, and second using a modified K-M method to account for inter-atomic interference (Supplementary Fig. 8). For the standard K-M scaling method, an estimate of the atomic inventory of the unit cell was used to calculate the theoretical coherent and incoherent scattering for independent atoms. The theoretical scattering calculation included 290 water molecules, 6 nitrate ions, and 1 lysozyme molecule in the unit cell, for a total of 1546 H, 613 C, 199 N, 493 O, and 10 S atoms. The total number of electrons was $Z = 10720$. Then, $I_{total,predicted}$ was calculated using Equation 40 in the Supplementary Methods, integrating over the observed region of reciprocal space. The modified K-M method was identical to the standard K-M method, except that $I_{total,predicted}$ was modified to include the interference between all atom pairs whose average inter-atomic distance could be predicted from the chemical structure alone (i.e. the protein sequence and known structure of water and solutes). Both covalent bonding and torsional restraints were considered. Molecular coordinates were taken from the chemical component dictionary[58], and pair distances between atoms of adjacent amino acids in the sequence were calculated assuming a planar peptide bond with a bond length of 1.33 Å. This resulted in 7405 pair distances for lysozyme, 6 per nitrate molecule, and 3 per water molecule. Then, the following bonding correction was calculated and added to the elastic scattering in Equation 40 in the Supplementary Methods:

$$I_{bond}(s) = 2 \sum_{n<m} f_n(s) f_m(s) \frac{\sin(2\pi s\, r_{nm})}{2\pi s\, r_{nm}}, \tag{4}$$

where $f$ is the atomic scattering factor, the sum is over bonded atom pairs, and $r_{nm}$ is the inter-atomic distance.

With synthetic data, both methods converge to the expected value of 1 for $\alpha$ (Supplementary Fig. 8, left). However, the modified method converges much more quickly and provides a more accurate scale factor at the resolutions (~2 Å) that are typical for macromolecular crystallography.

**Generation of the final diffuse map.** A final map of the diffuse intensities $I_D$ was generated on a fine grid with 13, 11, and 11 subdivisions along the reciprocal unit cell vectors $\mathbf{a}^*$, $\mathbf{b}^*$, and $\mathbf{c}^*$, respectively. The resolution range of the map was 25–1.25 Å (scattering vector of 0.04–0.8 Å$^{-1}$). Voxels containing Bragg peaks were excluded. Geometric and background corrections were applied (Equation 32 in the Supplementary Methods), and redundant observations were merged using the scaling model derived from the coarse map, described above. Errors were estimated using Poisson statistics and propagated through the correction, scaling and merging steps. When merging, observations with intensities more than $5\sigma$ from the mean were flagged as outliers and excluded. Intensities were placed on an absolute

scale using the previously-determined scale factor $\alpha$, and the theoretical incoherent scattering was subtracted (Equation 36 in the Supplementary Methods). To calculate the correlation coefficient for random half-datasets, the unmerged observations were randomly assigned using an algorithm that gave approximately equal statistical weight to each half-dataset, and the half-datasets were merged separately. The final map includes the isotropic scattering component due to elastic scattering.

**All-atom molecular dynamics (MD) simulation.** Four all-atom MD simulations of triclinic lysozyme crystals were performed with 1, 27 ($3 \times 3 \times 3$), 125 ($5 \times 5 \times 5$), and 343 ($7 \times 7 \times 7$) unit cells, similar to a 12 unit-cell simulation described previously[59]. The simulation was prepared using the AMBER suite, version 18[60], using the ff14SB force field for the protein[61,62], the SPC/E model for water[63], and the general Amber force field (GAFF)[64] parameters for the nitrate ion. The simulation boxes had dimensions equal to integer multiples of the experimentally determined room-temperature unit cell from PDB ID 4lzt ($a = 27.24$ Å, $b = 31.87$ Å, $c = 34.23$ Å, $\alpha = 88.52°$, $\beta = 108.53°$, $\gamma = 111.89°$). The simulation was initialized with the measured protein atom coordinates from PDB ID 4lzt (using the "A" alternate conformer)[24], arranged in a supercell grid. Nine nitrate ions were added per unit cell to neutralize the charge, as well as 290 water molecules. The number of water molecules was manually adjusted in order to achieve ~1 atm pressure at 295 K, resulting in 290, 293, 284, and 270 waters per protein chain in the 1, 27, 125, and 343 unit cell simulations, respectively. The simulations were equilibrated for about 0.2 µs and continued for an additional 5, 5, 2, and 1 µs, respectively, saving coordinates every 0.4 ns. A time step of 4 fs was used, where non-water hydrogen masses are set to 3 amu, with a corresponding decrease in the mass of its bonded atom[65].

For each snapshot, structure factors were calculated from the atomic coordinates using the ccp4[52] program sfall with its default grid parameters, a resolution of 0.95 Å, and a VDWR parameter of 3.0. The B-factor was set to 15 Å$^2$ for each atom. This value for a "snapshot" B-factor smooths the electron density distribution to allow the Fourier transforms used by sfall to obtain a converged result; this was tested by comparing to test calculations using twice as many grid points in each dimension and for test calculations in which the "snapshot" B-factor was varied between 5 and 20 Å$^2$. Since every atom was assigned the same B-factor, its effect can be undone by multiplying the structure factors coming from the sfall run by $\exp(+Bs^2/4)$. The Bragg intensity per unit cell (Equation 16 in the Supplementary Methods) then was calculated using $I_B = N^{-2} \langle F(h_0) \rangle^2$ where $F(h_0)$ is the supercell structure factor evaluated at the Bragg positions $h_0$, $N$ is the number of unit cells, and brackets represent an average over all saved simulation frames (see Section 1 of the Supplementary Methods). Similarly, the diffuse scattering per unit cell (Equation 15 in the Supplementary Methods) was calculated using $I_D = N^{-1}(\langle F^2 \rangle - \langle F \rangle^2)$. The whole procedure is encapsulated in the md2diffuse.sh script, distributed as a part of the AmberTools distribution (http://ambermd.org).

**Lattice dynamics simulation.** Lattice dynamics simulations and model refinement were performed in Matlab. Protein molecules were modeled as rigid bodies, and the lattice contacts were modeled as an elastic network[13,37,38,66,67] with pair-wise interactions between $\alpha$ carbons. The lattice contacts were identified in the all-atom structure determined in this study (PDB ID 6o2h). First, atoms with alternate conformers were assigned to their occupancy-weighted average positions. Then, the atomic coordinates of the 26 nearest neighbors in the lattice were generated by applying the crystal symmetry operators. A lattice contact was defined between any atom in the central protein chain that came within 4 Å of an atom belonging to a neighbor. Finally, the network was reduced to a $C_\alpha$ model. If any atoms belonging to a pair of residues formed a lattice contact, a spring was created between the $C_\alpha$ atoms in the network. A total of 100 intermolecular springs were modeled, of which 50 were unique due to crystal symmetry.

Two types of pair potential were modeled: Gaussian and directional, as follows:

$$V_{jj'}^{(Gauss.)} = \frac{1}{2} \gamma_{jj'} \left| \mathbf{u}_{(j)} - \mathbf{u}_{(j')} \right|^2 \tag{5}$$

and

$$V_{jj'}^{(dir.)} = \frac{1}{2} \gamma_{jj'} \left( (\mathbf{u}_{(j)} - \mathbf{u}_{(j')}) \cdot \hat{\mathbf{r}}_{(jj')} \right)^2, \tag{6}$$

where $j$ and $j'$ are the node indices, $\mathbf{u}$ is the displacement vector of a node from its equilibrium position, $\gamma$ is a spring constant, and $\hat{\mathbf{r}}_{(jj')}$ is the unit vector pointing from node $j$ to $j'$.

The equations of motion were solved in a rigid-body vibrational coordinate system using the Born/Von-Karman method (Section 3 of the Supplementary Methods). The diffuse scattering was calculated for a $13 \times 11 \times 11$ periodic supercell, chosen to match the level of detail in the experimental map, using the one-phonon approximation (Equation 62 in the Supplementary Methods). Terms in the one-phonon structure factor (Equation 63 in the Supplementary Methods) were calculated using the fast-Fourier transform-based method[68] with form factors approximated by four Gaussians and a constant[69,70]. The scattering contribution from the mean solvent density was modeled using Babinet's principle: since any constant can be added to the electron density without changing the structure factor (except at $\mathbf{s} = 0$), a constant density of $\rho_{solv.}$ surrounding a protein can be equivalently modeled by a density of 0 and $-\rho_{solv.}$ in the solvent-excluded region.

For reasons of computational convenience, the excluded solvent can then be represented by pseudo-atoms with Gaussian form factors. To calculate the Babinet representation, excluded voxels of the solvent mask from REFMAC5[54] were divided among the modeled atoms based on proximity. The constant mask density associated with each atom was approximated by a three-dimensional anisotropic Gaussian with the same first and second moments. The overall solvent scaling parameters $k_{solv.}$ and $B_{solv.}$ were then adjusted to minimize the least-squares difference between $F_{obs.}$ and $|F_{model}|$, defined as follows:

$$F_{model} = F_{calc.} + k_{solv.} \exp(-B_{solv.} s^2/4) \, F_{solv.}, \tag{7}$$

where $F_{calc.}$ is the structure factor of the modeled atoms (protein and ordered solvent) and $F_{solv.}$ is the structure factor of the excluded solvent. The refined parameters $k_{solv.}$ and $B_{solv.}$ were then applied to the excluded-solvent form factors. The resulting pseudo-atoms were included in the list of atoms occupying the unit cell and assigned to the same rigid group as the nearest protein atom.

Spring constants in the model were refined in order to minimize the least-squares difference between the simulated one-phonon scattering and the measured variational scattering around the 400 most intense halos in the 2–2.5 Å resolution range. Although the limited resolution range was used for refinement, the agreement of the model was ultimately assessed throughout reciprocal space (Fig. 2d). The reduced $\chi^2$ for refinement was calculated as follows:

$$\chi_{red.}^2 = \left( \sum_{n=1}^{N} M_n \right)^{-1} \sum_{n=1}^{N} \sum_{m=1}^{M_n} \left( \frac{I_{n,m}^{(meas.)} - I_{n,m}^{(calc.)} - b_n}{\sigma_{n,m}} \right)^2, \tag{8}$$

where $N = 400$ is the number of halos fit, $M_n$ is the number of measured voxels around the $n^{th}$ halo (typically $M_n = 13 \times 11 \times 11 - 1 = 1572$), $b_n$ is an arbitrary constant offset for each halo (determined separately by least-squares minimization for each $n$), and $\sigma$ is the experimental uncertainty. The spring constants were refined in four stages. In the first stage, all springs were set to Gaussian springs and assigned the same spring constant. In the second stage, springs belonging to the same interface (those involving a particular neighbor) were given the same spring constant. In the third stage, the pair-potentials for each interface were allowed to be a linear combination of Gaussian and directional. Finally, each pair potential was refined individually with a linear combination of Gaussian and directional springs. The overall $\chi^2$ was monitored during refinement to assess whether adding the extra degrees of freedom to the model significantly improved the fit (Supplementary Fig. 13).

After refining the model, the scattering was calculated throughout reciprocal space using the one-phonon approximation (Equation 62 in the Supplementary Methods). The Pearson correlation coefficient (CC) between the measured variational scattering map and the simulation was calculated within shells of constant resolution spanning 0.04 Å$^{-1}$ to 0.80 Å$^{-1}$ with a constant width of $\Delta s = 0.02$ Å$^{-1}$. Within each resolution bin, CC was calculated as follows:

$$CC = \frac{\sum_n (I_{meas.}(s_n) - \bar{I}_{meas.})(I_{calc.}(s_n) - \bar{I}_{calc.})}{\sqrt{\sum_n (I_{meas.}(s_n) - \bar{I}_{meas.})^2 \sum_n (I_{calc.}(s_n) - \bar{I}_{calc.})^2}}, \tag{9}$$

where $\bar{I}_{meas.}$ and $\bar{I}_{calc.}$ are the mean intensities in that resolution bin, and the sums are over all measured voxels within the resolution bin (Supplementary Fig. 14). For comparison with the MD simulation, which was calculated on a coarser $7 \times 7 \times 7$ sub-sampled reciprocal lattice, the full map was interpolated at the voxels of the coarser grid by least-squares fitting a 2nd order polynomial over all neighboring voxels (a $3 \times 3 \times 3$ kernel). Voxels at the Bragg positions (those with integer Miller indices) were excluded. The CC was calculated, as above, between the interpolated simulated map and a similarly interpolated experimental map (Fig. 2d).

**Internal protein dynamics simulation.** Internal dynamics simulations and model refinement were performed in Matlab. The dynamics of lysozyme within the crystal environment were simulated using an all-atom elastic network where each residue was restrained to move as a rigid body, and lattice contacts were explicitly modeled. To generate the model, first the atoms with alternate conformers were assigned to their occupancy-weighted average positions. Then, springs were created between any pair of non-H protein atoms belonging to different residues within a cutoff distance of 4 Å. Intermolecular springs were modeled between atoms in the protein and those of its neighbors in the lattice within the 4 Å cutoff distance. All springs were of the directional type (Eq. (6)).

The equations of motion for a single unit cell were solved using the Born/Von-Karman method as described for the lattice dynamics simulation, above, except that the potential energy function was modified in order to remove those modes associated with rigid-body motion of the entire protein. This was done by assigning the component of displacement associated with such motions a restoring force of zero. The normal modes associated with rigid-body displacements then have eigenvalues of zero and are eliminated during generalized inversion of the dynamical matrix (discussed in Section 3.3 of the Supplementary Methods). More specifically, components of the Hessian matrix (Equation 45 in the Supplementary Methods) were modified as follows:

$$\mathbf{V}_{(l,l')} := \mathbf{P}^T \mathbf{V}_{(l,l')} \mathbf{P}, \tag{10}$$

where $\mathbf{P}$ is an operator that projects out the rigid-body component of displacement,

$$\mathbf{P} = \mathbf{I} - (\mathbf{A} \backslash \mathbf{A}_0)(\mathbf{A}_0 \backslash \mathbf{A}), \tag{11}$$

$\mathbf{I}$ is a $6m \times 6m$ identity matrix ($m = 129$ is the number of residues), $\mathbf{A}$ is a $3n \times 6m$ matrix ($n$ is the number of atoms in the protein) that transforms between the Cartesian atomic displacement coordinates, $\mathbf{u}$, and the generalized coordinates of the internal dynamics model (Equation 43 in the Supplementary Methods), $\mathbf{A}_0$ is a $3n \times 6$ matrix that transforms between $\mathbf{u}$ and the generalized coordinates of the lattice dynamics model, and the forward slash signifies left matrix division (if $\mathbf{X} = \mathbf{A} \backslash \mathbf{A}_0$, then $\mathbf{X}$ is the least squares solution to the system of equations $\mathbf{AX} = \mathbf{A}_0$).

The model was parameterized with one coupling constant per residue, so that a spring connecting a pair of atoms ($j$ and $j'$) has a spring constant equal to the geometric mean of the residues' coupling constants $g_i$ and $g_{i'}$, as follows:

$$\gamma_{j,j'} = \sqrt{g_i g_{i'}}. \tag{12}$$

The parameters were optimized in order to minimize the $\chi^2$ between the measured and simulated atomic displacement parameters (ADPs), calculated as follows:

$$\chi^2 = \sum_{j=1}^{N} \sum_{n=1}^{9} \left( \left( \mathbf{U}_j^{(meas.)} \right)_n - \left( \mathbf{U}_j^{(latt.)} + \mathbf{U}_j^{(int.)} \right)_n \right)^2, \tag{13}$$

where $\left( \mathbf{U}_j \right)_n$ is the $n^{th}$ component of the ADP for atom $j$ ($\mathbf{U}_j$ is a symmetric $3 \times 3$ matrix with 9 components), $\mathbf{U}_j^{(latt.)}$ is the calculated ADP for the fully-refined lattice dynamics model, and $\mathbf{U}_j^{(int.)}$ is the calculated ADP for the internal dynamics model (Equation 56 in the Supplementary Methods).

After refining the model, the displacement correlations were assessed using the directional correlation coefficient, defined as follows:

$$CC_{j,j'} = \frac{\hat{\mathbf{r}}_{j,j'}^T \left\langle \mathbf{u}_j \mathbf{u}_{j'}^T \right\rangle \hat{\mathbf{r}}_{j,j'}}{\sqrt{(Tr\mathbf{U}_j/3)(Tr\mathbf{U}_{j'}/3)}}, \tag{14}$$

where $\hat{\mathbf{r}}_{j,j'}$ is the unit vector pointing from atom $j$ to $j'$.

We also defined an alternate model of internal protein motion where the modes associated with rigid-body displacements of individual domains are suppressed. Residues were assigned to three groups[45] as follows: 5–36 and 98–129 to $\alpha$, 40–94 to $\beta$, and those remaining to the hinge region. The Hessian matrix was modified as described above, except that the $\mathbf{P}$ operator appearing in Eq. (10) was calculated as follows:

$$\mathbf{P} = \mathbf{I} - (\mathbf{A} \backslash \mathbf{A}_1)(\mathbf{A}_1 \backslash \mathbf{A}), \tag{15}$$

where $\mathbf{I}$ is a $6m \times 6m$ identity matrix ($m = 129$ is the number of residues), $\mathbf{A}_1$ is a $3n \times 6d$ matrix ($d = 3$ is the number of groups) that projects from the generalized coordinate system of the 3-group model to the atomic displacements (Equation 43 in the Supplementary Methods). The model was parameterized and refined as described above for the unrestrained model, and the directional correlation was calculated using Eq. (14).

**Diffuse Patterson map calculation.** Diffuse Patterson maps were calculated in Matlab as the Fourier transform of the diffuse scattering (Section 1.3 in the Supplementary Methods), using a three-dimensional fast-Fourier transform (FFT).

The experimental diffuse map was pre-processed before performing the FFT to compensate for missing data. First, missing voxels in the diffuse map were filled in with the mean values from neighboring voxels. Then, the mean intensity in each resolution shell was subtracted, and voxels beyond the resolution limit of the map were filled with zeros. Finally, the data array were zero-padded to yield a diffuse Patterson map with a real-space voxel approximately 0.3 Å on a side (the voxel dimensions were $a/91$, $b/107$, and $c/115$, where $a$, $b$, and $c$ are the lattice constants).

The diffuse Patterson map for the refined vibrational model (lattice + internal) was calculated without approximation in the central region where $r < 25$ Å. To perform the calculation efficiently, the scattering per unit cell (Equation 59 in the Supplementary Methods) was rearranged to single out a reference unit cell ($l = 0$):

$$I_D = \sum_j f_j \left\{ \sum_{l',j'} f_{j'} e^{2\pi i \mathbf{s} \cdot (\mathbf{r}_j - \mathbf{r}_{j'} + \mathbf{r}_0)} T_j T_{j'} \left( T_{0j,l'j'} - 1 \right) \right\}, \tag{16}$$

where the first sum runs over all atoms in the unit cell, $f_i$ is the atomic scattering factor, $T_j$ is the Debye-Waller factor (Equation 60 in the Supplementary Methods), and $T_{0j,l'j'}$ depends on the cross-terms of the covariance matrix (Equation 61 in the Supplementary Methods with $l = 0$). The term in the curly brackets resembles the standard structure factor equation for the primed atoms, except that the origin is shifted and the Debye-Waller factor is replaced by

$$(T_{eff})_{0j,l'j'} = T_j T_{j'} \left( T_{0j,l'j'} - 1 \right). \tag{17}$$

The effective Debye-Waller factor was separated into contributions from lattice and

internal motion:

$$T_{\text{eff}} = T_{\text{eff}}^{\text{latt}} + T_{\text{eff}}^{\text{int}}. \qquad (18)$$

The lattice term was calculated as follows:

$$\left(T_{\text{eff}}^{\text{latt}}\right)_{0j,l'j'} = T_j T_{j'} \left(T_{0j,l'j'}^{\text{latt}} - 1\right), \qquad (19)$$

where the experimentally determined ADPs were used in $T_j$ and $T_{j'}$, and $T_{0j,l'j'}^{\text{latt}}$ was calculated from the refined lattice model (Equations 55 and 61 in the Supplementary Methods). This corresponds to the definition used in the one-phonon simulation (Equation 62 in the Supplementary Methods). For the internal motions, the effective Debye-Waller factor was calculated as follows:

$$\left(T_{\text{eff}}^{\text{int}}\right)_{0j,l'j'} = T_j^{\text{latt}} T_j^{\text{int}} T_{j'}^{\text{latt}} T_{j'}^{\text{int}} T_{0j,l'j'}^{\text{latt}} \left(T_{0j,l'j'}^{\text{int}} - 1\right), \qquad (20)$$

where the $T$'s are calculated from the covariance matrices of the lattice and internal dynamics simulations.

Excluded-solvent effects were modeled by pseudo-atoms with Gaussian scattering factors, as described above for the lattice dynamics simulation. In Eq. (16), terms in curly brackets were calculated using the FFT-based method as described for the lattice dynamics simulation, except that each atom had an effective Debye-Waller factor (Eq. (18)) and coordinates relative to $\mathbf{r}_j$. Since only the central part of the Patterson was desired, the sum was carried out over all atoms in the unit cell and its 26 nearest neighbors that satisfied $|\mathbf{r}_j - \mathbf{r}_{j'} - \mathbf{r}_{l'} + \mathbf{r}_0| < 29$ Å (the cutoff distance was chosen to be somewhat larger than the maximum distance of 25 Å to avoid truncation artifacts). After calculating the diffuse intensity map using Eq. 16, the mean intensity in each resolution shell was subtracted and voxels outside the experimental resolution limit of 1.25 Å were set to zero. Then, the map was zero-padded, and the Patterson function was calculated using the three-dimensional FFT, as described above for the experimental map.

The reciprocal space correlation coefficients between diffuse Patterson maps were also calculated in Matlab. First, real space voxels with $|\mathbf{r}| < 2$ Å or $|\mathbf{r}| > 25$ Å were set to zero, and maps were truncated at $|x| < a$, $|y| < b$ and $|z| < c$ so that the reciprocal space map would be oversampled by a factor of 2 in each direction. Then, the inverse FFT of each truncated map was calculated. The Pearson correlation coefficients (Eq. (9)) between the experimental and simulated intensity maps were calculated in shells of constant resolution spanning 0.04 to 0.80 Å$^{-1}$ with bin widths of $\Delta s = 0.04$ Å$^{-1}$.

## Data availability

The atomic coordinates and structure factors have been deposited in the Protein Data Bank under accession code 6o2h. Diffraction images have been deposited in the SBGrid Data Bank under ID 747 (https://doi.org/10.15785/SBGRID/747). The processed diffuse maps have been deposited the Coherent X-ray Imaging Data Bank under ID 128 (https://doi.org/10.11577/1601281). Other data are available from the corresponding author upon reasonable request.

## Code availability

The source code used in this study for reciprocal space mapping and scattering simulation are publicly available on GitHub (https://github.com/ando-lab/mdx-lib/tree/natcomm). Also included are scripts for processing the lysozyme dataset and for fitting the lattice dynamics model and elastic network models. Software used for structure determination was curated by SBGrid[71].

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

## Acknowledgements

We thank staff at the Cornell High Energy Synchrotron Source (CHESS) and MacCHESS beamline F1 for supporting diffraction data collection, W.C. Thomas for assistance with data collection, V.D. Pillar for useful discussions, and S.M. Gruner, W.C. Thomas, M.B. Watkins, B.R. Crane, and A.S. Byer for critical reading of the paper. CHESS is supported by NSF Grant DMR-1332208, and the MacCHESS facility is supported by NIH/NIGMS Grant GM-103485. This work was supported by NIH Grants GM117757 (to S.P.M.), GM100008 (to N.A.), GM124847 (to N.A.), and GM122086 (to D.A.C.) and by start-up funds from Princeton University and Cornell University (to N.A.).

## Author contributions

S.P.M. performed crystallization, data collection, and structure determination with assistance from N.A., D.A.C. performed and analyzed the MD simulations. S.P.M. developed methods for processing and analyzing diffuse scattering data and performed vibrational simulations. The paper was written by S.P.M. and N.A. and edited by all authors. N.A. conceived of the experiments and coordinated the research.

## Competing interests

The authors declare no competing interests.
