## [Peer Review File · Nature Communications]

Reviewers' comments:

Reviewer #1 (Remarks to the Author):

This paper demonstrates a key way that diffuse scattering can be used to move beyond the traditional description of atomic motions in protein crystallography. The traditional description uses temperature factors, which are not sensitive to atom pair correlations. The temperature factors correspond to the diagonal elements of the atom displacement covariance matrix; a natural extension therefore is to add the off-diagonal elements. Previous studies modeling the covariance matrix using crystal vibrations demonstrated limited agreement with diffuse scattering data (see, esp. Refs. 13 and 23, and also Ref. 19). This study distinguishes itself by using a high-quality, finely sampled data set obtained using a modern detector and light source, introducing a number of innovative data processing techniques, and using a hierarchical approach to modeling crystal vibrations. The strong, sharply varying intensity in the neighborhood of the Bragg peaks is first modeled using rigid protein units connected to neighbors by ~ 100 springs. The spring constants are refined to optimize the agreement with the sharp diffuse features. The cloudy intensity further from the Bragg peaks is then modeled using a model of internal dynamics, where a network of springs is generated using 129 additional local elastic parameters, one per residue. The elastic parameters are refined to optimize the agreement with the cloudy diffuse features. The resulting model provides a substantially more accurate description of the rich pattern of anisotropic intensity than previous models of protein diffuse scattering.

The advances in the methods and the insights gained are very impressive and will substantially influence the future direction of the field. In particular the scaling model in Eq. (1) used to merge the data, and the use of a modified integral method of Krogh-Moe to place the map on an absolute scale of electron units per unit cell should be influential for data processing. Regarding the crystal vibrations model, the method to remove the potential energy corresponding to the projection along rigid body motions appears to be novel and is interesting. In addition there is a MD study presented that is distinguished by the large number of unit cells in the model and very strong agreement with the isotropic component of the diffuse data, which can be a dominant part of the signal for protein crystal diffraction. This model does not agree as well with the fine-grained diffuse features as well as the vibrational model; however, there are no free parameters, and agreement seems substantial enough to create opportunities for improving the MD simulations.

The paper illustrates by example a way forward for transforming the way dynamics are treated in protein crystallography. It therefore has far-reaching implications in the way crystal structures might be used in the future across diverse disciplines and applications.

Specific comments:

p. 2. "In particular, the diffuse pattern is typically a small oscillation on top of a large background and is therefore easily corrupted by intense Bragg peaks." Consider replacing "oscillation" with something like "variation" to make sure that readers don't imagine that the signal typically looks like a wave.

p. 3. "This map reveals, for the first time, a surprisingly large contribution of long-ranged correlated motions across multiple unit cells, while also enabling detection of protein motions in a manner that is consistent with both Bragg diffraction and diffuse scattering." Consider softening the "for the first time" language as elements of this have been demonstrated in some form (e.g. Ref. 9).

p. 4. "The diffuse scattering is dominated by a broad, isotropic scattering ring with a peak at 3 \AA° (Fig. 1C, ii) that arises from short-ranged disorder generally attributed to water." Consider revising this sentence to not create an impression that the ring is purely due to water, as the MD suggests that the isotropic ring is a combination of water and protein, which was also observed in, e.g., Ref. 29.

p. 4. "The presence of such clear halo scattering was unexpected as it implies that there are correlations between atoms in different unit cells." Was this completely unexpected? Halos and long-range correlations were studied, e.g., in Refs. 9, 19, and 21.

p. 5. "The characteristic exponent of approximately -2 is also found for other intense halos throughout the map (Fig. 2A, right)." Fig. 2A only shows the distribution of exponents for the 100 most intense halos. Although the mode of the distributions is -2, the exponent appears to vary substantially, mostly between -2 and -3. Does this imply that not all peaks are equally well-modeled using acoustic modes? Given how important this paper will be in establishing the future direction of the field, I recommend that the authors consider emphasizing more any ways in which their model doesn't perfectly describe the data, so readers can understand what needs to be done next. Also the authors appear to be in a position to determine whether the halos are better described using an exponent of -2 or using a shape corresponding to exponential real-space correlations, which can yield sharper power-law decays (e.g., compare Eqs. 3 and 4 in Ref. 15). While such a comparison might be beyond the scope of this paper, it would be an interesting subject for future work.

p. 6. "Although this may seem to be a small motion, the intense halo signal is derived from the collective motions of many proteins." Consider explaining how constructive interference on such a long length scale can increase the local intensity values, which might help complete the thought here for some readers.

p. 6. "Moreover, there is a significant gap between CC (Fig. 2D, orange) and CC* (Fig. 2D, black dashed), which estimates the maximum CC a model can achieve, given the precision of the data [36]." Although CC* is probably a good measure to use, note that the interpretation of CC* isn't necessarily the same for diffuse and Bragg data, given that diffuse measurements nearby in reciprocal space can be highly correlated. It might be good to discuss this at least briefly so that readers can better understand how to think about the CC* and CC_{1/2} statistics for diffuse scattering data.

p. 8. "Strikingly, the lattice dynamics model clearly outperforms all-atom MD (Fig. 2C-D)." Even though it might be clear from the context, perhaps remind the readers that this assessment specifically applies to the halos. After all the performance of the MD is remarkably good in modeling the isotropic intensity, and presumably substantially better in recovering this part of the signal than any other model considered here.

p. 9. "In order to model only the internal protein dynamics, the Hessian matrix describing the restoring forces was modified to suppress rigid-body motion of the entire protein." Did the authors attempt to simultaneously refine spring constants for internal and lattice dynamics using the halos and cloudy features? Are there any comparative advantages of the hierarchical approach adopted?

p. 9. "To assess the importance of hinge-bending in the model, we examined the covariance matrices C_{ij} for all alpha carbon pairs and calculated a "directional correlation", which is the component of C_{ij} along the inter-atomic vector normalized by the r.m.s. displacements of the two atoms (Methods)." Some other studies have used the trace of each 3x3 atom pair submatrix and normalized by the rms displacements. Is there a specific reason to use the directional correlation, or was this definition chosen arbitrarily?

p. 9. "At large distances, the experimental diffuse Patterson displays peaks at the lattice nodes as expected (Fig. 5A, left), whereas continuous features are most intense at short distances (Fig. 5A, right)." Consider a brief comment on why is this expected, for uneducated readers.

p. 12. "A background dataset was collected for each crystal by translating the sample out of the beam along the spindle axis and collecting 1 s exposures while rotating at 1 degree per sec." It

looks like the backgrounds were collected every 1 deg whereas the crystal exposures were collected every 0.1 degree. How were the backgrounds specifically used? Were they interpolated before subtraction from each individual crystal exposure?

pp. 13-18. Processing of diffuse data. It is difficult to follow here whether the final diffuse map has had the isotropic component subtracted. Perhaps clarify this either at the beginning or end of this section.

p. 20 "Spring constants in the model were refined in order to minimize the least-squares difference between the simulated one-phonon scattering and the measured variational scattering around the 400 most intense halos in the 2-2.5 Å resolution range." Consider also mentioning here that, although the limited resolution range was used for refinement, the agreement of the model was ultimately assessed throughout reciprocal space (Fig. 2D). Was the halo shape of the model also compared to the data outside of this limited resolution range?

p. 21. "The dynamics of lysozyme within the crystal environment were simulated using an all-atom elastic network where each residue was restrained to move as a rigid body, and lattice contacts were explicitly modeled." The dynamical model used presumably allows these rigid units translate along with the alpha-carbon, but not to rotate. This probably limits the agreement of individual ADPs of all heavy atoms between this model and the crystal structure refined against the Bragg data alone. It would be good to address this issue in the manuscript and perhaps shed some light on the path forward to obtain more realistic models that simultaneously agree with the Bragg and diffuse data.

p. 25. "The atomic coordinates and structure factors have been deposited in the Protein Data Bank under accession code 6o2h. Diffraction images have been deposited in the SBCGrid Data Bank. All other data are available in the main text or the supplementary materials." Will the processed 3D data and matlab scripts for the data processing be made available as well?

Michael Wall

Reviewer #2 (Remarks to the Author):

This paper represents a major breakthrough in macromolecular diffuse scatter (DS) data collection, processing and modelling. It will definitely have a strong influence on the field. The algorithms and protocols for achieving the high quality diffuse data are creative, thorough, and novel and will no doubt change the way others process and analyze their diffuse scatter data. The refinement of the lattice model against the DS data is not entirely novel as elastic network models have been used before, but it is still innovative, very clearly described, and a remarkably good fit.

A major claim of the work is that two alternative dynamic models that are indistinguishable in the Bragg data are distinguishable using the DS data. This has been a long sought-after achievement in the field, and potentially makes this a landmark work. The evidence supporting this claim, however, is a bit weak. Only about 0.01 difference in correlation coefficient. The consistency of this increase across resolution bins makes it more convincing, but it is at best just crossing over the cusp of discriminatory utility. Nevertheless, the paper that crosses over the cusp is what makes it a landmark.

A problem with the paper is that this finding is quite well hidden. It took me a while to find it. And until I did find it I was having a hard time deciding what the paper was about. One phrase: "two models that fit the ADPs equally well" is key, and should perhaps be backed up with two numbers that evaluate the fit to the ADPs of

each model. My biggest question mark about this (Fig 5H) is that the best-matching model also has more fit-able parameters. It is seldom unexpected that more parameters leads to a better fit. I understand that the model is more sophisticated than that, but the "parameter count" will be the first impression of most readers. A better control would have been a model with the same number of parameters, perhaps constraining two equal-sized regions that are not actual domains.

I suspect the reason for "burying" this key finding is because the authors share my healthy skepticism about it, but I don't think that is an advisable strategy. Important results like this should be front-and-center and the discussion of them balanced. Do not try to hide it just because it is a close call. As long as you are honest about alternative interpretations the reader will be happy to reach their own conclusion. This is why the exact agreement with the ADPs is important to mention.

A related specific comment, the text:

"Residues were assigned to three domain..."

seems inconsistent. Aren't there only two domains? alpha and beta?

It is also noteworthy that there is high correlation between the halos and the Bragg peaks. This should not be relegated to the SI. It is very important as it affects the interpretation of the entire work. For example, the fitting of the elastic network model to the 400 top halo intensities is no doubt essentially equivalent to fitting them to the 400 top Bragg peaks. It would be fair to say that the elastic network was not fitted to the diffuse data at all! That is a valuable argument against bias. Some might even argue that the halos are not even "diffuse" since they are localized features. This correlation of the halos to the Bragg intensities is a known problem in the DS field, and one of the many great things about this paper is that the authors present a new way of dealing with them. The correlation to Bragg intensity is therefore very important to bring forward.

As a matter of fact, a significant concern that I had on my first read is the large number of parameters involved in these elastic network models as well as in all the correction factors for data processing. I found myself repeatedly concerned about over-fitting and looking for discussion of how it was evaluated. Why not set aside a handful of halos as a "free R"? Or, better yet, why not refine the models against the Bragg data? Why 400 and not all of them? Or a random set? I suspect the protocols presented were not the only ones tried, but rather the ones that worked the best. There is nothing intrinsically wrong with this, but a very brief description about what didn't work as well will be very valuable to the rest of the field.

XDS - the body of corrections applied in data processing was indeed extensive, but XDS uses a very similar set of corrections in processing the Bragg data. I found myself very curious to see if the XDS absorption corrections, for example, matched those used for the DS. This may be something for the SI, but if they do match that should not only simplify future analysis a great deal, but also validate the approach described here.

It is the understanding of this reviewer that the proper name for these halo features is Huang scattering, but there are no references to the work of Huang, the seminal papers of Kanzaki in the 1950s, or even a review. It would probably be a good idea to check that body of work for relevance here?

Babinet solvent. Although Babinet bulk solvent modelling has long been a feature available in REFMAC5, the implementation here appears rather unconventional and the description of it a bit cryptic. For example, REFMAC5 does not write out an "excluded solvent mask", it writes out a bulk solvent mask. Do you mean the Babinet inverse of the bulk solvent mask? If so, that is not clear. It also took me a while to interpret "same first and second moments of density" which density? the atom? Or the bulk solvent map? Was the anisotropic Gaussian centered on the atom, or the centroid of the "density"? I am assuming this solvent model was implemented because it made a

difference, so describing it more clearly is warranted.

Difference Pattersons: I found it very instructive to go through the SI and see all the excellent controls that were done. But something I was expecting since the abstract and was surprised to find missing was a figure showing the residual Patterson features after subtracting the calculated Patterson. Such a difference Patterson ought to be very useful for doing exactly what this work set out to do: remove "uninteresting" DS features such as the halos and lattice dynamics and reveal what is left: internal dynamics. Was this difference Patterson calculated and found to be uninformative? If so, then that is perhaps worth knowing too.

Absolute scale:

These authors seem to have a different idea than others about what "absolute scale" means. Some might regard the photon scale as the "absolute scale" of the real world. In that case the Pilatus measures on absolute scale. It might be more informative to say "absolute electron scale", or may be just "electron scale". Why not simply scale the Bragg structure factors to those calculated from the refined model? F_{calc} is always on an absolute electron scale. This would allow recovery of the required scale factor needed to put DS on an absolute scale: the product of illuminated volume and incident flux density.

One final specific comment:

S37 is not Parseval's Theorem. Also, "s" cannot go to infinity because of the finite size of the Ewald sphere. Integrating the form factor over 4π , however, does indeed recover the tabulated total elastic scattering cross section. However, the authors are right that the extremely high angles where S37 applies are at too high angle to be observed. Scaling the Bragg data may perhaps be more accurate than extrapolating from high angle? At the very least, it is a cross-check.

Oh, and finally, publication in this journal is highly recommended. DS is an old field, but recently interest has grown enormously. This will be a landmark DS paper.

Reviewer #3 (Remarks to the Author):

The major goal of this paper is to use diffuse scattering data to inform models of collective protein motions. This is a landmark paper that unites many disparate observations in the field and pushes the state of the art forward much more so than any paper since Wall et al, 1997 PNAS.

Through careful data collection, the authors are able to separate Bragg and diffuse scattering. The major experimental advance over previous work is that their fine-scale analysis enables them to integrate diffuse halos surrounding the Bragg peaks. This data yields the observations needed to model lattice dynamics. They find that lattice dynamics explain a significant fraction of the diffuse scattering data. Nonetheless, the authors noticed residual B-factors and turned to internal protein motions to explain the remaining disorder, which leaves signals both around the Bragg peaks and in hazy streaks and clouds between them.

To explain these residual features, they tested both normal modes analysis (NMA) and full molecular dynamics (MD). Furthermore, they were able to use Patterson analysis to choose between redundant NMA models, conquering an outstanding challenge in the field of macromolecular diffuse scattering. Surprisingly, the NM model that accounts for lattice motions and internal protein motions matches the data better than a crystalline MD model. What does this mean for MD that a reduced representation fits better?

Overall, the data collection and processing are extremely thorough. Opening up these analytical methods to the community is the next step - and publishing their code is the only essential revision we would request prior to publication.

Despite our enthusiastically positive interpretation, we do have a few minor questions and requests for clarification:

While examining the exponential decay in halos around the Bragg peaks, why are the 100 most intense peaks between 2 Å and 10 Å focused on? In Figure 2 it appears that there is a skew in the distribution of exponents toward a sharper decay ($n > 2$). How do the histograms look when more halos are sampled? Is it possible that this sharp decay could be explained by Bragg peaks that are leaking into adjacent voxels?

The authors are rigorous and explicit in their modeling efforts and make impressive strides forward. Still, we are left with questions about these models. For refinement of the lattice dynamics model, a small fraction of halos were chosen. Why did the authors not use all the halos? Why was the angular range of 2 Å to 2.5 Å chosen for refinement? Why does this resolution range differ from the analysis of halo decay (2 Å to 10 Å)?

As we commented above, we were surprised to see that a NMA model matched the diffuse intensities better than a crystalline MD model. We wonder whether incorporating the isotropic component of the diffuse scatter would alter this interpretation? Furthermore, since the authors scrupulously subtracted sources of isotropic background scatter, why was the remaining isotropic portion of diffuse scattering not used for refinement of the NMA and MD models?

Using diffuse scattering data to distinguish between competing models of motion has been a longstanding challenge in the field of macromolecular diffuse scattering, and we are impressed with the authors' work in this regard. This is really a breakthrough! We were surprised to see how subtle the effects of restraining domain motions were upon the Δ PDF in Figure S17, can the authors comment on the statistical significance of this difference? What is the uncertainty in the Patterson map, and how does this play into the interpretation of the best model?

We have no major stylistic recommendations. The figures are elegant and clearly represent the main points of the paper. Similarly, the text is clear and concise, with thorough expansion in the supplemental material.

On a final note, this paper pushes the field forward, and we believe there is room for further speculation. A few areas to consider:

How might crystallographers who encounter more mosaic Bragg peaks (these are some of the least mosaic crystals in existence!) separate the Bragg signal from the diffuse signal to analyze halos?

In what ways can NMA models and MD be further improved to match diffuse scattering data? What complications might arise in crystals with more complex unit cells, and how can this be overcome?

How do they reconcile the results of ref 18 with their analysis of the lattice dynamics (different systems obviously)?

The authors have done an excellent job of carefully collecting data, thoroughly analyzing it, and clearly explaining their work. We think that digging into the questions above may add to the already substantial impact of this paper, and look forward to their replies. Nonetheless, we think this important paper is worthy of publication as is (noting the caveat of code release).

We review non-anonymously, James Fraser and Alex Wolff (UCSF)

We thank all reviewers for their positive comments, which were received on December 31, 2019. Our revised manuscript includes links to the processed data as well as the code. Suggested changes to the text, which were quite minor, are tracked in color. Supplementary Figure 16 has also been annotated with R^2 values.

Reviewer #1

This paper demonstrates a key way that diffuse scattering can be used to move beyond the traditional description of atomic motions in protein crystallography. The traditional description uses temperature factors, which are not sensitive to atom pair correlations. The temperature factors correspond to the diagonal elements of the atom displacement covariance matrix; a natural extension therefore is to add the off-diagonal elements. Previous studies modeling the covariance matrix using crystal vibrations demonstrated limited agreement with diffuse scattering data (see, esp. Refs. 13 and 23, and also Ref. 19). This study distinguishes itself by using a high-quality, finely sampled data set obtained using a modern detector and light source, introducing a number of innovative data processing techniques, and using a hierarchical approach to modeling crystal vibrations. The strong, sharply varying intensity in the neighborhood of the Bragg peaks is first modeled using rigid protein units connected to neighbors by ~100 springs. The spring constants are refined to optimize the agreement with the sharp diffuse features. The cloudy intensity further from the Bragg peaks is then modeled using a model of internal dynamics, where a network of springs is generated using 129 additional local elastic parameters, one per residue. The elastic parameters are refined to optimize the agreement with the cloudy diffuse features. The resulting model provides a substantially more accurate description of the rich pattern of anisotropic intensity than previous models of protein diffuse scattering.

The advances in the methods and the insights gained are very impressive and will substantially influence the future direction of the field. In particular the scaling model in Eq. (1) used to merge the data, and the use of a modified integral method of Krogh-Moe to place the map on an absolute scale of electron units per unit cell should be influential for data processing. Regarding the crystal vibrations model, the method to remove the potential energy corresponding to the projection along rigid body motions appears to be novel and is interesting. In addition there is a MD study presented that is distinguished by the large number of unit cells in the model and very strong agreement with the isotropic component of the diffuse data, which can be a dominant part of the signal for protein crystal diffraction. This model does not agree as well with the fine-grained diffuse features as well as the vibrational model; however, there are no free parameters, and agreement seems substantial enough to create opportunities for improving the MD simulations.

The paper illustrates by example a way forward for transforming the way dynamics are treated in protein crystallography. It therefore has far-reaching implications in the way crystal structures might be used in the future across diverse disciplines and applications.

We thank the reviewer for his thorough reading and perceptive summary.

Specific comments:

p. 2. "In particular, the diffuse pattern is typically a small oscillation on top of a large background and is therefore easily corrupted by intense Bragg peaks." Consider replacing "oscillation" with something like "variation" to make sure that readers don't imagine that the signal typically looks like a wave.

Thank you for the suggestion. We changed "oscillation" to "variation" in the revised manuscript (p. 2).

p. 3. "This map reveals, for the first time, a surprisingly large contribution of long-ranged correlated motions across multiple unit cells, while also enabling detection of protein motions in a manner that is consistent with

Friday, January 17, 2020

both Bragg diffraction and diffuse scattering.” Consider softening the “for the first time” language as elements of this have been demonstrated in some form (e.g. Ref. 9).

While statements of novelty are sometimes discouraged in scientific papers, here we feel that it will help the reader distinguish our work from previous studies. Long-ranged correlations have indeed been studied before, however our diffuse map represents the first truly quantitative measurement with sufficient detail to determine the contribution of lattice disorder to both diffuse scattering and crystallographic B-factors. In ref. 9, for example, the X-ray film was overexposed, so there was the dynamic range was insufficient to actually observe the decay of the halos. Halo width was therefore used as a proxy for intensity, and this apparently led the authors to underestimate the contribution of lattice motion to the B-factors.

p. 4. “The diffuse scattering is dominated by a broad, isotropic scattering ring with a peak at 3 Å° (Fig. 1C, ii) that arises from short-ranged disorder generally attributed to water.” Consider revising this sentence to not create an impression that the ring is purely due to water, as the MD suggests that the isotropic ring is a combination of water and protein, which was also observed in, e.g., Ref. 29.

We agree with the reviewer. The text now reads:

“Although this ring is generally attributed to water, short-ranged protein disorder also contributes.” (p. 4)

p. 4. “The presence of such clear halo scattering was unexpected as it implies that there are correlations between atoms in different unit cells.” Was this completely unexpected? Halos and long-range correlations were studied, e.g., in Refs. 9, 19, and 21.

In the past, either relatively “soft” correlations were fit to the halos (refs. 9, 19) or the halos were studied indirectly far from the Bragg peaks (ref. 21). We have changed the wording to be more precise about what was unexpected, as follows:

“The presence of such intense halo scattering near the Bragg peaks was unexpected, as it implies the correlations between atoms in different unit cells are significant and long-ranged.” (p. 4)

p. 5. “The characteristic exponent of approximately -2 is also found for other intense halos throughout the map (Fig. 2A, right).” Fig. 2A only shows the distribution of exponents for the 100 most intense halos. Although the mode of the distributions is -2, the exponent appears to vary substantially, mostly between -2 and -3. Does this imply that not all peaks are equally well-modeled using acoustic modes? Given how important this paper will be in establishing the future direction of the field, I recommend that the authors consider emphasizing more any ways in which their model doesn’t perfectly describe the data, so readers can understand what needs to be done next. Also the authors appear to be in a position to determine whether the halos are better described using an exponent of -2 or using a shape corresponding to exponential real-space correlations, which can yield sharper power-law decays (e.g., compare Eqs. 3 and 4 in Ref. 15). While such a comparison might be beyond the scope of this paper, it would be an interesting subject for future work.

We refrained from discussing this comparison with prior models because the isotropic power-law description is only approximate, and we quickly move on to a much better model (lattice dynamics) that fully captures the anisotropy of the halos (Fig. 3C) and their modulation by the molecular transform (Supplementary Fig. 10). That said, you are right that we are in a position to compare the simple vibrational model with the liquid-like correlations introduced by Caspar and colleagues. For the interested reader, we include such a comparison below in our response, which will be published alongside the manuscript:

Equation 3 in Ref. 15 has a power law of -4, which is a much steeper decay than we observe, so it can be rejected straight away. Equation 4 in Ref 15 approaches the vibrational model (power law of -2) when the correlation length goes to infinity. If the correlation length is small, it predicts a deviation from power-law behavior (a Gaussian-like roll-off) as one approaches the Bragg peak, which we do not observe for triclinic lysozyme. Thus we can say that *if* a characteristic correlation length does exist for this crystal, it must be well beyond the limits of our measurement (much greater than ~ 300 Å).

p. 6. "Although this may seem to be a small motion, the intense halo signal is derived from the collective motions of many proteins." Consider explaining how constructive interference on such a long length scale can increase the local intensity values, which might help complete the thought here for some readers.

Thank you for the suggestion. We modified the sentence to read: "[...], the intense halo signal is derived from constructive interference of scattered radiation from many proteins moving collectively" (p. 6).

p. 6. "Moreover, there is a significant gap between CC (Fig. 2D, orange) and CC (Fig. 2D, black dashed), which estimates the maximum CC a model can achieve, given the precision of the data [36]." Although CC* is probably a good measure to use, note that the interpretation of CC* isn't necessarily the same for diffuse and Bragg data, given that diffuse measurements nearby in reciprocal space can be highly correlated. It might be good to discuss this at least briefly so that readers can better understand how to think about the CC* and CC1/2 statistics for diffuse scattering data.*

The interpretation of CC is somewhat different between Bragg and diffuse scattering data because in the latter case the signal-to-noise ratio depends on how the map is sampled (compare Fig. 2D and Supplementary Fig. 14), whereas this is not a consideration with Bragg data because sampling is fixed. However, we are not aware of any fundamental limitation of CC* for signals that contain measurements that are correlated nearby in reciprocal space, assuming those correlations are part of the signal and do not arise from systematic errors.

p. 8. "Strikingly, the lattice dynamics model clearly outperforms all-atom MD (Fig. 2C-D)." Even though it might be clear from the context, perhaps remind the readers that this assessment specifically applies to the halos. After all the performance of the MD is remarkably good in modeling the isotropic intensity, and presumably substantially better in recovering this part of the signal than any other model considered here.

Thank you for pointing out this opportunity to clarify the text. It now reads: "Strikingly, the lattice dynamics model clearly outperforms all-atom MD in its ability to describe the variational component (Fig. 2C-D)." (p. 8).

p. 9. "In order to model only the internal protein dynamics, the Hessian matrix describing the restoring forces was modified to suppress rigid-body motion of the entire protein." Did the authors attempt to simultaneously refine spring constants for internal and lattice dynamics using the halos and cloudy features? Are there any comparative advantages of the hierarchical approach adopted?

Note that we did not refine the internal dynamics model using diffuse data (only ADPs were used). We agree that global fitting (to diffuse, Bragg, etc.) could yield an even better fit and allow more information to be extracted. We did not attempt this because calculating diffuse scattering from the internal motions model is too slow to use in a refinement cycle (the one-phonon approximation is not accurate enough for internal motions, and the full Patterson calculation is comparatively laborious). However, these are merely computational issues and can hopefully be overcome in future work.

p. 9. "To assess the importance of hinge-bending in the model, we examined the covariance matrices C_{ij} for all alpha carbon pairs and calculated a "directional correlation", which is the component of C_{ij} along the interatomic vector normalized by the r.m.s. displacements of the two atoms (Methods)." Some other studies have used

Friday, January 17, 2020

the trace of each 3x3 atom pair submatrix and normalized by the rms displacements. Is there a specific reason to use the directional correlation, or was this definition chosen arbitrarily?

We introduced the directional correlation because it emphasizes the type of motion (co-linear translation) that matters most for diffuse scattering, especially in the context of domain motions. If the trace of the submatrix is used instead, then atoms on opposite ends of a rigid-body appear highly correlated due to rotations of the body, however the diffuse scattering from this kind of correlation is pretty minimal.

p. 9. "At large distances, the experimental diffuse Patterson displays peaks at the lattice nodes as expected (Fig. 5A, left), whereas continuous features are most intense at short distances (Fig. 5A, right)." Consider a brief comment on why is this expected, for uneducated readers.

Thank you for the suggestion. The text now reads "[...] peaks at the lattice nodes, as expected (the Fourier transform of a lattice is also a lattice), whereas [...]" (p. 10).

p. 12. "A background dataset was collected for each crystal by translating the sample out of the beam along the spindle axis and collecting 1 s exposures while rotating at 1 degree per sec." It looks like the backgrounds were collected every 1 deg whereas the crystal exposures were collected every 0.1 degree. How were the backgrounds specifically used? Were they interpolated before subtraction from each individual crystal exposure?

Since the backgrounds varied gradually with spindle angle, interpolation was not necessary. Instead, the same 1-degree background image was used for all 10 of the 0.1-degree crystal exposures collected over the same range. This means that a single background observation is typically counted multiple times when integrating a voxel. The number of times each background pixel contributes was taken into account during error propagation.

pp. 13-18. Processing of diffuse data. It is difficult to follow here whether the final diffuse map has had the isotropic component subtracted. Perhaps clarify this either at the beginning or end of this section.

We added this sentence at the end of the section: "The final map includes the isotropic scattering component due to elastic scattering." (p. 18).

p. 20 "Spring constants in the model were refined in order to minimize the least-squares difference between the simulated one-phonon scattering and the measured variational scattering around the 400 most intense halos in the 2-2.5 Å resolution range." Consider also mentioning here that, although the limited resolution range was used for refinement, the agreement of the model was ultimately assessed throughout reciprocal space (Fig. 2D). Was the halo shape of the model also compared to the data outside of this limited resolution range?

Thank you for the suggestion. We incorporated this statement into the text (p. 21). The anisotropy parameter was calculated only within the range used for fitting.

p. 21. "The dynamics of lysozyme within the crystal environment were simulated using an all-atom elastic network where each residue was restrained to move as a rigid body, and lattice contacts were explicitly modeled." The dynamical model used presumably allows these rigid units translate along with the alpha-carbon, but not to rotate. This probably limits the agreement of individual ADPs of all heavy atoms between this model and the crystal structure refined against the Bragg data alone. It would be good to address this issue in the manuscript and perhaps shed some light on the path forward to obtain more realistic models that simultaneously agree with the Bragg and diffuse data.

Actually, rotations are allowed in this model - the degrees of freedom are equivalent to a TLS model in which every residue is its own TLS group. It does not reproduce the refined ADPs exactly because in REFMAC the

Friday, January 17, 2020

ADPs were refined for individual atoms (TLS was not used). Furthermore, the parameterization of one coupling constant per residue was perhaps overly conservative and also limited agreement. As you suggest, a more flexible and detailed model might be developed to fit ADPs exactly while being physically motivated and having a simple parameterization. However, this is a tall order indeed! We expect it will be a topic of ongoing discussion for the field.

p. 25. "The atomic coordinates and structure factors have been deposited in the Protein Data Bank under accession code 6o2h. Diffraction images have been deposited in the SBGrid Data Bank. All other data are available in the main text or the supplementary materials." Will the processed 3D data and matlab scripts for the data processing be made available as well?

We will deposit the processed 3D data in the CXIDB (www.cxidb.org), and the code will be open source and available on GitHub (<https://www.github.com/ando-lab/>) at the time of publication. We have added the following to the Data Availability section:

"The processed diffuse map has been deposited the Coherent X-ray Imaging Data Bank under ID 128." (p. 26)

We have added the following Code Availability statement:

"The source code used in this study for reciprocal space mapping and scattering simulation are publicly available on GitHub. Also included are scripts for processing the lysozyme dataset and for fitting the lattice dynamics model and elastic network models." (p. 26)

Michael Wall

We thank Michael Wall for his thorough review and for suggesting many improvements to the manuscript.

Reviewer #2

This paper represents a major breakthrough in macromolecular diffuse scatter (DS) data collection, processing and modelling. It will definitely have a strong influence on the field. The algorithms and protocols for achieving the high quality diffuse data are creative, thorough, and novel and will no doubt change the way others process and analyze their diffuse scatter data. The refinement of the lattice model against the DS data is not entirely novel as elastic network models have been used before, but it is still innovative, very clearly described, and a remarkably good fit.

Although similar lattice models have been applied in the small molecule literature [for example, Welberry et al, Metallurg. Mater. Trans. A 43, 1434 (2012)], to our knowledge, our application to protein diffuse scattering is novel. Note that previous studies that used elastic networks for proteins did not model correlations between unit cells in the crystal lattice (e.g. refs. 17 and 13).

A major claim of the work is that two alternative dynamic models that are indistinguishable in the Bragg data are distinguishable using the DS data. This has been a long sought-after achievement in the field, and potentially makes this a landmark work. The evidence supporting this claim, however, is a bit weak. Only about 0.01 difference in correlation coefficient. The consistency of this increase across resolution bins makes it more convincing, but it is at best just crossing over the cusp of discriminatory utility. Nevertheless, the paper that crosses over the cusp is what makes it a landmark.

The ability of diffuse scattering to distinguish between alternate models is indeed a key application. Several recent papers have claimed to do this already (e.g. refs. 17, 29, 20), however the model/data correlations were not con-

Friday, January 17, 2020

vincing (lattice dynamics were not modeled explicitly in the references above, which likely contributed to the poor correlations). We believe the landmark achievement of our paper is the ability to measure and model the total scattering to high accuracy. Although we also show the ability to discriminate between models by gain in correlation coefficient, this is meant to demonstrate feasibility and suggest a future direction. More work is required to do this in a robust way (i.e. it must be shown to be repeatable for different crystals, to not depend too much on how the model is parameterized, or on the details of the lattice dynamics fit). For these reasons, we chose to not make this the main point of the paper. However, we agree that it represents a landmark in the field, and we anticipate that it will be recognized as such.

A problem with the paper is that this finding is quite well hidden. It took me a while to find it. And until I did find it I was having a hard time deciding what the paper was about. One phrase:

"two models that fit the ADPs equally well"

is key, and should perhaps be backed up with two numbers that evaluate the fit to the ADPs of each model.

Supplementary Fig. 16 now includes R^2 values in each panel (coefficient of determination) to quantify agreement between measured and modeled ADPs. This statistical analysis does not change our conclusions (the R^2 values for the two models are the same to two significant figures).

My biggest question mark about this (Fig 5H) is that the best-matching model also has more fit-able parameters. It is seldom unexpected that more parameters leads to a better fit. I understand that the model is more sophisticated than that, but the "parameter count" will be the first impression of most readers. A better control would have been a model with the same number of parameters, perhaps constraining two equal-sized regions that are not actual domains.

To clarify, the two models have the same number of parameters (129 - one for each residue). Furthermore, the models are refined to the ADPs, not the diffuse scattering. Thus, the notion that "more parameters leads to a better fit" does not apply in this case. The diffuse scattering provides an independent check on the model. We have repeated that $m = 129$ in the definition on p. 24.

I suspect the reason for "burying" this key finding is because the authors share my healthy skepticism about it, but I don't think that is an advisable strategy. Important results like this should be front-and-center and the discussion of them balanced. Do not try to hide it just because it is a close call. As long as you are honest about alternative interpretations the reader will be happy to reach their own conclusion. This is why the exact agreement with the ADPs is important to mention.

See above.

A related specific comment, the text:

"Residues were assigned to three domain..."

seems inconsistent. Aren't there only two domains? alpha and beta?

Thank you for catching the ambiguous use of "domains". We have changed "domains" to "groups" in this context. The reason there are three groups and not two is that there is also a "hinge region" in addition to the alpha and beta domains, as discussed on p. 24 and in Ref. 45.

It is also noteworthy that there is high correlation between the halos and the Bragg peaks. This should not be relegated to the SI. it is very important as it affects the interpretation of the entire work. For example, the fitting of the elastic network model to the 400 top halo intensities is no doubt essentially equivalent to fitting them to the 400 top Bragg peaks. It would be fair to say that the elastic network was not fitted to the diffuse data at all! That is a valuable argument against bias. Some might even argue that the halos are not even "diffuse" since they are

Friday, January 17, 2020

localized features. This correlation of the halos to the Bragg intensities is a known problem in the DS field, and one of the many great things about this paper is that the authors present a new way of dealing with them. The correlation to Bragg intensity is therefore very important to bring forward.

Note that we did not fit the elastic network model to the integrated halo *intensities* (i.e. 400 data points). Rather, the model was fit to the ALL of the variational intensity around 400 halos (400*1572 voxels ~ 600,000 data points) – see p. 7. Crucially, the information contained is very different. Mainly, we are fitting the variations in halo anisotropy with direction in reciprocal space, not the relative intensities of the halos (as shown in Fig. 3B-C and S13). Note also that the Bragg peak intensities have already been taken into account, because the refined structure is used to calculate the molecular transform (see Supplementary Fig. 10 and Eq. S62-S64).

As a matter of fact, a significant concern that I had on my first read is the large number of parameters involved in these elastic network models as well as in all the correction factors for data processing. I found myself repeatedly concerned about over-fitting and looking for discussion of how it was evaluated. Why not set aside a handful of halos as a "free R"? Or, better yet, why not refine the models against the Bragg data? Why 400 and not all of them? Or a random set? I suspect the protocols presented were not the only ones tried, but rather the ones that worked the best. There is nothing intrinsically wrong with this, but a very brief description about what didn't work as well will be very valuable to the rest of the field.

We share your concern about overfitting in general. However, it is not warranted in this case. Although 400 halos were used in refinement, representing ~1 percent of the available data, the model was evaluated using all of the variational scattering (Fig 2C-D). As suggested by Reviewer 1, this is now also stated in the Methods (on p. 21). Furthermore, the model is strongly overdetermined by the data (see our response to a similar question by reviewer 3, below).

XDS - the body of corrections applied in data processing was indeed extensive, but XDS uses a very similar set of corrections in processing the Bragg data. I found myself very curious to see if the XDS absorption corrections, for example, matched those used for the DS. This may be something for the SI, but if they do match that should not only simplify future analysis a great deal, but also validate the approach described here.

Note that we disabled scaling in XDS and used aimless instead (however the scaling models are similar between the two programs). A crucial difference between Bragg and diffuse scaling is the treatment of radiation damage (i.e. B-factor decay). We found that the scale factors from aimless do not cleanly separate effects of intensity decay due to radiation damage from other effects (e.g. absorption), so they could not be used reliably for diffuse scattering. Clearly there is a need for a scaling program to use both Bragg and diffuse data. However, that is beyond the scope of this study.

It is the understanding of this reviewer that the proper name for these halo features is Huang scattering, but there are no references to the work of Huang, the seminal papers of Kanzaki in the 1950s, or even a review. It would probably be a good idea to check that body of work for relevance here?

Huang scattering refers to the diffuse pattern resulting from static defects in the crystal lattice and is often identified by its characteristic “butterfly”-shaped halo intensity (see for example: Krogstad et al. Nat. Materials, 2018, 17(8), 718). This is not to be confused with the dynamic (vibrational) theory of Born and Huang (Ref. 39) - same Huang, different theory. A survey of the literature on scattering theory is beyond the scope of the paper, but the essential references can be found in Section 3 of the Supplementary Methods, and within the thorough review by Eckold referenced in the Main Text (Ref. 40).

Babinet solvent. Although Babinet bulk solvent modelling has long been a feature available in REFMAC5, the implementation here appears rather unconventional and the description of it a bit cryptic. For example, REF-

Friday, January 17, 2020

MAC5 does not write out an "excluded solvent mask", it writes out a bulk solvent mask. Do you mean the Babinet inverse of the bulk solvent mask? If so, that is not clear.

REFMAC5 employs a binary solvent mask which is zero for excluded solvent and 1 for bulk solvent. The excluded solvent mask refers to the part of the mask that is zero. We have modified the text to make this clear (see below).

It also took me a while to interpret "same first and second moments of density" which density? the atom? Or the bulk solvent map? Was the anisotropic Gaussian centered on the atom, or the centroid of the "density"?

The “density” refers to the mask (the density is 1 in the excluded region, and 0 in the bulk region). The Gaussians are centered at the first moment of the mask density as stated, not at the atomic positions. We modified the text to clarify these points:

“To calculate the Babinet representation, excluded voxels of the solvent mask from REFMAC5 were divided among the modeled atoms based on proximity. The constant mask density associated with each atom was approximated by a three-dimensional anisotropic Gaussian with the same first and second moments.” (p. 20).

I am assuming this solvent model was implemented because it made a difference, so describing it more clearly is warranted.

The reason we employ the unusual solvent model is for computational convenience. The goal is to get as close as possible to the REFMAC result without using a discrete real-space grid. We did not use a traditional Babinet solvent mask in REFMAC5 because it is considered a poor choice for refinement of high-resolution data.

Difference Pattersons: I found it very instructive to go through the SI and see all the excellent controls that were done. But something I was expecting since the abstract and was surprised to find missing was a figure showing the residual Patterson features after subtracting the calculated Patterson. Such a difference Patterson ought to be very useful for doing exactly what this work set out to do: remove "uninteresting" DS features such as the halos and lattice dynamics and reveal what is left: internal dynamics. Was this difference Patterson calculated and found to be uninformative? If so, then that is perhaps worth knowing too.

Early on in this project, we explored using the difference Patterson to evaluate the internal motions model. However, we quickly abandoned this approach. Features in the difference Patterson are not necessarily due to internal motions; they also reflect subtle errors in the lattice dynamics model, and especially errors in the calculated molecular transform. Therefore, we felt that emphasizing the difference Patterson was likely to cause confusion. Instead, we adopted the more conservative approach of evaluating the fit of the total model (lattice + internal) to the data.

Absolute scale:

These authors seem to have a different idea than others about what "absolute scale" means. Some might regard the photon scale as the "absolute scale" of the real world. In that case the Pilatus measures on absolute scale. It might be more informative to say "absolute electron scale", or may be just "electron scale". Why not simply scale the Bragg structure factors to those calculated from the refined model? F_{calc} is always on an absolute electron scale. This would allow recovery of the required scale factor needed to put DS on an absolute scale: the product of illuminated volume and incident flux density.

The term “absolute scale” is defined in the first mention (p. 4) as “absolute scale of electron units per unit cell”. This is the standard meaning of “absolute scale” in the scattering field.

One final specific comment: S37 is not Parseval's Theorem.

From the preceding text it is clear that S37 was *derived* using Parseval's theorem: "Application of Parseval's theorem from harmonic analysis allows us to write the integral [...] as follows:". (p. 23 of SI).

Also, "s" cannot go to infinity because of the finite size of the Ewald sphere. Integrating the form factor over 4π , however, does indeed recover the tabulated total elastic scattering cross section. However, the authors are right that the extremely high angles where S37 applies are at too high angle to be observed. Scaling the Bragg data may perhaps be more accurate than extrapolating from high angle? At the very least, it is a cross-check.

To clarify, integration of S37 over the Ewald sphere would indeed yield the total elastic cross section, however this is not what is done in K-M scaling. The "s" variable here is not confined to the Ewald sphere. Regarding the second question, the Bragg data are not on an absolute scale, so we do not see how they could be used for scaling in this case.

Oh, and finally, publication in this journal is highly recommended. DS is an old field, but recently interest has grown enormously. This will be a landmark DS paper.

We thank the reviewer for the engaging discussion. We hope our work will advance the field and help channel the recent interest in a productive direction.

Reviewer #3

The major goal of this paper is to use diffuse scattering data to inform models of collective protein motions. This is a landmark paper that unites many disparate observations in the field and pushes the state of the art forward much more so than any paper since Wall et al, 1997 PNAS.

Through careful data collection, the authors are able to separate Bragg and diffuse scattering. The major experimental advance over previous work is that their fine-scale analysis enables them to integrate diffuse halos surrounding the Bragg peaks. This data yields the observations needed to model lattice dynamics. They find that lattice dynamics explain a significant fraction of the diffuse scattering data. Nonetheless, the authors noticed residual B-factors and turned to internal protein motions to explain the remaining disorder, which leaves signals both around the Bragg peaks and in hazy streaks and clouds between them.

To explain these residual features, they tested both normal modes analysis (NMA) and full molecular dynamics (MD). Furthermore, they were able to use Patterson analysis to choose between redundant NMA models, conquering an outstanding challenge in the field of macromolecular diffuse scattering. Surprisingly, the NM model that accounts for lattice motions and internal protein motions matches the data better than a crystalline MD model. What does this mean for MD that a reduced representation fits better?

We very much appreciate the well-written and insightful summary, which was also posted as a comment on our preprint (<https://doi.org/10.1101/805424>).

Regarding the fit to MD, this was commented on briefly in the Manuscript and in more detail in the supporting information (Supplementary Fig. 12): "In particular, the accuracy of MD for diffuse scattering appears to be limited by errors in the average electron density (Supplementary Fig. 12)." As can be seen in Supplementary Fig. 12C, the predicted Bragg intensities are actually quite different from the experimentally observed ones; this is largely because the average atomic coordinates in MD differ slightly from those determined experimentally (see Ref. 59). Note that MD was not restrained by the experiment: it is a prediction, not a fit. Since the halo intensities are strongly correlated with the nearby Bragg intensities (Supplementary Fig. 12A-B), it is not surprising that MD

Friday, January 17, 2020

has difficulty reproducing the diffuse scattering at high resolution. The “reduced representation” (lattice dynamics model) benefits from using the experimentally-determined atomic coordinates in the calculation (i.e. it gets the Bragg data right by construction).

Overall, the data collection and processing are extremely thorough. Opening up these analytical methods to the community is the next step - and publishing their code is the only essential revision we would request prior to publication.

The code for data processing and scattering simulation will be open source and available on GitHub (see response to Reviewer 1).

Despite our enthusiastically positive interpretation, we do have a few minor questions and requests for clarification:

While examining the exponential decay in halos around the Bragg peaks, why are the 100 most intense peaks between 2 Å and 10 Å focused on?

The most intense peaks in this resolution range are focused on here because they have the best signal-to-noise. The halo intensity spans about 2 orders of magnitude (Fig. 2A), so a strong signal is needed in order to observe the power law decay far from the Bragg peak.

In Figure 2 it appears that there is a skew in the distribution of exponents toward a sharper decay ($n > 2$). How do the histograms look when more halos are sampled? Is it possible that this sharp decay could be explained by Bragg peaks that are leaking into adjacent voxels?

The sharper decay is not due to Bragg peaks leaking in. We were conservative with our choice of voxel size to avoid this issue (see Supplementary Fig. 3). The reason not all of the halos have the expected exponent of -2 is because the theory is only approximate, and does not include modulation by the molecular transform (see Supplementary Fig. 10). See our response to a related question from Reviewer 1.

The authors are rigorous and explicit in their modeling efforts and make impressive strides forward. Still, we are left with questions about these models. For refinement of the lattice dynamics model, a small fraction of halos were chosen. Why did the authors not use all the halos?

A small fraction of the halos was used for refining the lattice model because in theory relatively little data is required. In materials science, for example, where the unit cells are small and relatively few intense Bragg peaks are observed, there is typically enough information in a handful of halos to determine the 21 independent elastic constants. Thus, the use of 400 halos is actually highly redundant. The reason for using more than necessary is to minimize the effect of errors in the measurement and in the molecular transform (i.e. errors in the atomic coordinates).

Why was the angular range of 2 Å to 2.5 Å chosen for refinement? Why does this resolution range differ from the analysis of halo decay (2 Å to 10 Å)?

Halo anisotropies do not depend on $|s|$ in the one-phonon approximation, so fitting to a single resolution shell was adequate. The particular resolution range of 2 - 2.5 Å was chosen because it has a large number of intense halos that uniformly sample directions in reciprocal space (points in Fig. 3C).

Friday, January 17, 2020

As we commented above, we were surprised to see that a NMA model matched the diffuse intensities better than a crystalline MD model. We wonder whether incorporating the isotropic component of the diffuse scatter would alter this interpretation?

The result would be the same if CC is the metric for agreement, because the mean is subtracted from each resolution shell when calculating the covariances (Eq. 9). While it is true that MD does an excellent job with the isotropic component (Supplementary Fig. 9), it is not appropriate to evaluate normal-modes models using the isotropic intensity because they do not include bulk solvent correlations (by rough estimation, bulk solvent is responsible for half of the isotropic scattering).

Furthermore, since the authors scrupulously subtracted sources of isotropic background scatter, why was the remaining isotropic portion of diffuse scattering not used for refinement of the NMA and MD models?

See above. Also, note that MD is a prediction. MD models were not refined.

Using diffuse scattering data to distinguish between competing models of motion has been a longstanding challenge in the field of macromolecular diffuse scattering, and we are impressed with the authors' work in this regard. This is really a breakthrough! We were surprised to see how subtle the effects of restraining domain motions were upon the Δ PDF in Figure S17, can the authors comment on the statistical significance of this difference? What is the uncertainty in the Patterson map, and how does this play into the interpretation of the best model?

The experimental uncertainty in the central part of the Patterson is very small, and can be quantified by CC* in reciprocal space (dashed line in Fig. 5G). For the models, the Patterson calculation is essentially exact: numerical errors are insignificant on the scale of Supplementary Fig. 17. So the gap between CC* and CC in Fig. 5 is indeed significant.

We have no major stylistic recommendations. The figures are elegant and clearly represent the main points of the paper. Similarly, the text is clear and concise, with thorough expansion in the supplemental material.

On a final note, this paper pushes the field forward, and we believe there is room for further speculation. A few areas to consider:

How might crystallographers who encounter more mosaic Bragg peaks (these are some of the least mosaic crystals in existence!) separate the Bragg signal from the diffuse signal to analyze halos?

In what ways can NMA models and MD be further improved to match diffuse scattering data?

What complications might arise in crystals with more complex unit cells, and how can this be overcome?

How do they reconcile the results of ref 18 with their analysis of the lattice dynamics (different systems obviously)?

These are excellent questions. We plan to address many of them specifically in future work.

The authors have done an excellent job of carefully collecting data, thoroughly analyzing it, and clearly explaining their work. We think that digging into the questions above may add to the already substantial impact of this paper, and look forward to their replies. Nonetheless, we think this important paper is worthy of publication as is (noting the caveat of code release).

We review non-anonymously, James Fraser and Alex Wolff (UCSF)

We thank James Fraser and Alex Wolff for the thoughtful review.

REVIEWERS' COMMENTS:

Reviewer #1 (Remarks to the Author):

The authors have done an excellent job of addressing my original comments. I'm glad that I was able to make some useful suggestions and am honored to have been able to help in some small way.

Michael Wall

Reviewer #2 (Remarks to the Author):

I believe the authors have understood my first round of comments with one exception:

"Bragg data are not on an absolute scale"

Yes they are, if they are calculated. I was suggesting scaling to F_{calc} for Bragg data. This does not impact the present work, however, so no further response or modification is required.

Overall, the authors seem confident in disagreeing with their reviewers on several points, which is fine by me. That should not be a barrier to publication. What I do strongly suggest, however, is improving the clarity. The authors seem to be more concerned in general with addressing reviewer comments in the rebuttal text without making any changes to the manuscript that led to those comments and questions. No doubt the three reviewers will not be the only people to ever have these questions. Even if the rebuttal is made public it does not count as archival literature. The main text should stand on its own and clearly communicate the work. Specific examples that I must insist upon are:

"the models are refined to the ADPs, not the diffuse scattering"

This was in no way clear to this reviewer after carefully reading the main text the first time. The authors are advised to write this statement somewhere in the main text.

"the two models have the same number of parameters"

Yes, but they do not have the same number of restraints/constraints, so the number of "free parameters" is different, yes? This was my source of concern that I believe will be shared by most readers. Please clear this up, and do it in the main text.

Finally, and most important:

"two models that fit the ADPs equally well"

please back up this critical statement with the actual R numbers in the main text, not just in the SI.

That is all. Good job and good luck. I look forward to seeing this in print!

Reviewer #2

Overall, the authors seem confident in disagreeing with their reviewers on several points, which is fine by me. That should not be a barrier to publication. What I do strongly suggest, however, is improving the clarity. The authors seem to be more concerned in general with addressing reviewer comments in the rebuttal text without making any changes to the manuscript that led to those comments and questions. No doubt the three reviewers will not be the only people to ever have these questions. Even if the rebuttal is made public it does not count as archival literature. The main text should stand on its own and clearly communicate the work. Specific examples that I must insist upon are:

"the models are refined to the ADPs, not the diffuse scattering"

This was in no way clear to this reviewer after carefully reading the main text the first time. The authors are advised to write this statement somewhere in the main text.

This was already stated in several places: the section heading in Results (p. 7), the final sentence of the following paragraph (p. 7), and in the Methods (p. 18, equations 12 and 13). That said, we understand that the reader may be primed to think of fitting directly the diffuse scattering. As suggested, we added an additional reminder:

"In contrast, the diffuse Patterson calculated from the in internal motions model, refined to the residual ADPs, shows prominent fluctuations [...]" (p. 8).

"the two models have the same number of parameters"

Yes, but they do not have the same number of restraints/constraints, so the number of "free parameters" is different, yes? This was my source of concern that I believe will be shared by most readers. Please clear this up, and do it in the main text.

We have revised the final paragraph in Results and added a Supplementary Note to make this point more clearly (substantially altered text is underlined):

"This restrained model has the same number of free parameters as the unrestrained model (Supplementary Note 1) and it is also able to reproduce the experimentally derived B-factors ($R^2 = 0.88$ for both models, see Supplementary Fig. 16C). However, the internal dynamics are significantly different (Fig. 4D), underscoring the challenge of distinguishing differing models of protein motion from Bragg data alone. Yet, the two models are distinguishable by diffuse scattering: fluctuations in the diffuse Patterson decay more rapidly with domain motions suppressed (Supplementary Fig. 17), leading to a subtle but systematically worse CC, particularly at high resolution (Fig. 5H, dashed)." (p. 8)

A Supplementary Note was added to provide further clarification. This note addresses the reviewers' question concerning restraints and free parameters:

"Supplementary Note 1. Number of free parameters in the elastic network model. Two elastic network models for internal protein motion were fit to the residual ADPs. The number of normal modes for the elastic network is $129 \times 6 - 6 = 771$, since there are 129 rigid groups (one per residue) with 6 degrees of freedom each, and the restraint that there is no rigid-body motion of the entire protein reduces the number of modes by 6. In the model with domain motion suppressed, similar restraints applied to the 3 rigid groups (α , β , and hinge) result in $129 \times 6 - 3 \times 6 = 756$ normal modes. Thus, by restraining 3 groups instead of 1, the number of normal modes is very moderately reduced, which in principle might make it more difficult to fit the ADPs. However, we did not adjust the normal mode amplitudes to fit the ADPs as was done in prior studies [Riccardi et al 2010, Benschoten et al 2016].

Monday, February 3, 2020

Instead, we refined the physical model (spring constants) through a conservative parameterization with one coupling constant per residue (or 129 free parameters, see Equation 12 in the Main Text). Thus, the ability of each model to fit the ADPs is not limited by the number of normal modes, but is instead limited primarily by the parameterization, which is the same in both cases.”

Finally, and most important:

"two models that fit the ADPs equally well"

please back up this critical statement with the actual R numbers in the main text, not just in the SI.

The R^2 value now appears in the main text on p. 8 (see paragraph reproduced above).

We thank the reviewer for suggesting these ways of improving overall clarity.